# A near atomic structure of the active human apoptosome

Tat Cheung Cheng[1], Chuan Hong[2], Ildikó V Akey[1], Shujun Yuan[3], Christopher W Akey[1]*

[1]Department of Physiology and Biophysics, Boston University School of Medicine, Boston, United States; [2]Janelia Research Campus, Howard Hughes Medical Institute, Ashburn, United States; [3]Department of Biologics Research - Protein Sciences, U.S. Innovation Center, Bayer Healthcare, San Franciso, United States

**Abstract** In response to cell death signals, an active apoptosome is assembled from Apaf-1 and procaspase-9 (pc-9). Here we report a near atomic structure of the active human apoptosome determined by cryo-electron microscopy. The resulting model gives insights into cytochrome c binding, nucleotide exchange and conformational changes that drive assembly. During activation an acentric disk is formed on the central hub of the apoptosome. This disk contains four Apaf-1/pc-9 CARD pairs arranged in a shallow spiral with the fourth pc-9 CARD at lower occupancy. On average, Apaf-1 CARDs recruit 3 to 5 pc-9 molecules to the apoptosome and one catalytic domain may be parked on the hub, when an odd number of zymogens are bound. This suggests a stoichiometry of one or at most, two pc-9 dimers per active apoptosome. Thus, our structure provides a molecular framework to understand the role of the apoptosome in programmed cell death and disease.

## Introduction

*For correspondence: cakey@bu.edu

**Competing interests:** The authors declare that no competing interests exist.

Programmed cell death occurs during tissue development and homeostasis to remove superfluous cells or alternatively, during disease states to remove damaged or rapidly dividing cells (*Song and Stellar, 1999*; *Salvesen and Dixit, 1997*; *Green and Evan, 2002*; *Danial and Korsmeyer, 2004*; *Thompson, 1995*). In general, this process allows cellular components to be recycled without activating inflammatory pathways (*Green and Evan, 2002*; *Inohara and Nuñez, 2003*). In humans, intrinsic cell death signals converge on mitochondria to release pro-apoptotic factors through outer membrane pores formed by Bcl-2 family members (*Brunelle and Letai, 2009*; *Tait and Green, 2010*). These factors include cytochrome c, which up-regulates assembly of the Apoptotic protease activation factor-1 (Apaf-1) to form the apoptosome (*Li et al., 1997*). Other released factors, such as Smac/DIABLO and Omi/HtrA2 (reviewed in *Tait and Green, 2010*), are pro-apoptotic and interact with inhibitory proteins to unblock procaspases –9, –3 and –7. This multi-pronged pathway allows activation of procaspase-9 on the apoptosome and subsequent proteolytic activation of procaspase-3 and –7, which results in selective proteolysis of target proteins and cell death (*Bratton and Salvesen, 2010*).

Apaf-1 is a member of the AAA+ ATPase family, has seven domains and exists as an inactive monomer in the cytoplasm of healthy cells (*Danot et al., 2009*; *Leipe et al., 2004*). When released, cytochrome c binds to β-propellers in Apaf-1 and may trigger nucleotide exchange with ATP/dATP replacing bound ADP/dADP (*Liu et al., 1996*; *Li et al., 1997*; *Zou et al., 1997*, *1999*; *Hu et al., 1998*, *1999*). This leads to the assembly of a heptameric apoptosome (*Acehan et al., 2002*). Caspase recognition domains (CARDs) are present as N-terminal effector domains in both Apaf-1 and procaspase-9 (pc-9). Thus, CARD-CARD interactions target inactive pc-9 monomers to the

**eLife digest** An adult human loses around 50–70 billion cells every day via a process termed apoptosis. The term arises from the Greek word that describes leaves "falling off" a tree, and the process entails damaged or unwanted cells essentially committing suicide in a controlled manner. As such, apoptosis keeps the number of cells in tissues and organs in check. It also allows components of old cells to be recycled to make new ones.

In cells that are targeted to die, a protein called cytochrome c interacts with another protein, named Apaf-1, together with a nucleotide triphosphate molecule. These steps work in concert to trigger the assembly of the apoptosome: a large wheel-like complex that contains seven copies each of Apaf-1 and cytochrome c. The central hub of the wheel then recruits a specific protein-cutting enzyme, which once activated sets in motion a cascade of events that dismantle the cell from the inside out.

Cheng et al. now use an electron microscope to reveal the three-dimensional structure of the active human apoptosome, in enough detail to determine the positions of many of the amino acids that make up the complex. The three dimensional model provides new insights into how Apaf-1 changes shape as the complex assembles in the presence of cytochrome c and nucleotide triphosphate. Cheng et al. went on to reveal a disk-like structure made from the parts of four Apaf-1 proteins that interact with the protein-cutting enzymes. This disk forms a spiral that sits atop the central hub of the wheel-like apoptosome. Finally, the findings suggest that, although the wheel like complex has seven spokes, at any one time the active apoptosome may only need two (or at most four) copies of the protein-cutting enzyme to trigger the cascade of events that lead to cell death

In the future, emerging technologies may provide high enough resolution to visualize fine details of the interactions between cytochrome c and Apaf-1, and reveal more about the disk-like spiral as well. This in turn will give a better understanding of how the apoptosome assembles and how the protein-cutting enzyme becomes activated.

apoptosome (*Yuan et al., 2011a*; *Hu et al., 2014*). During this step, a CARD 'disk' is formed that sits above the central hub, and is attached by linkers to the nucleotide binding domain (NBD) (*Yuan et al., 2010*, *2011a*, *2013*). The nature of the disk is not known but a recent hypothetical model suggests that two CARD-CARD interfaces may be used to form a quasi-helical stack, akin to structures in Death Domain complexes (*Hu et al., 2014*). Local proximity of procaspase-9 monomers on the apoptosome leads to protease activation. However, the precise mechanism of pc-9 activation after recruitment to the apoptosome is still being debated. Catalytic domains may associate in solution or they may interact with the apoptosome to form an active protease (*Yuan et al., 2011a*, *Bratton and Salvesen, 2010*; *Boatright et al., 2003*; *Hu et al., 2014*; *Würstle and Rehm, 2014*).

We now present a near atomic structure of the active human apoptosome determined with cryo-EM at a global resolution of ~4.1 Å. This work provides a model of Apaf-1 in an extended conformation with bound dATP and shows monomer interactions within the apoptosome. Improved clarity reveals interactions of cytochrome c with β-propellers in the V-shaped sensor domain. For the first time, molecular details are provided of a tilted, acentric disk formed by N-terminal CARDs from four Apaf-1 and three or four pc-9 molecules. In brief, the disk contains four pairs of Apaf-1/pc-9 CARDs in one turn of a spiral with the fourth pc-9 CARD being present at lower occupancy. This arrangement creates a mismatch between the CARD spiral in the disk and the c7 symmetry of the platform. We also verified that a pc-9 catalytic domain may be parked on the central hub (*Yuan et al., 2011a*), adjacent to a pc-9 CARD in the disk. In addition, three Apaf-1 CARDs are excluded from the disk but may retain a limited ability to bind pc-9 molecules, depending upon steric constraints. Stoichiometry measurements have suggested that 2 to 5 pc-9 molecules may be recruited by Apaf-1 CARDs on the apoptosome (*Malladi et al., 2009*; *Yuan et al., 2011a*; *Hu et al., 2014*). Hence, at any instant one or two pc-9 dimers may be tethered to the apoptosome. Implications for Apaf-1 assembly and apoptosome function are discussed.

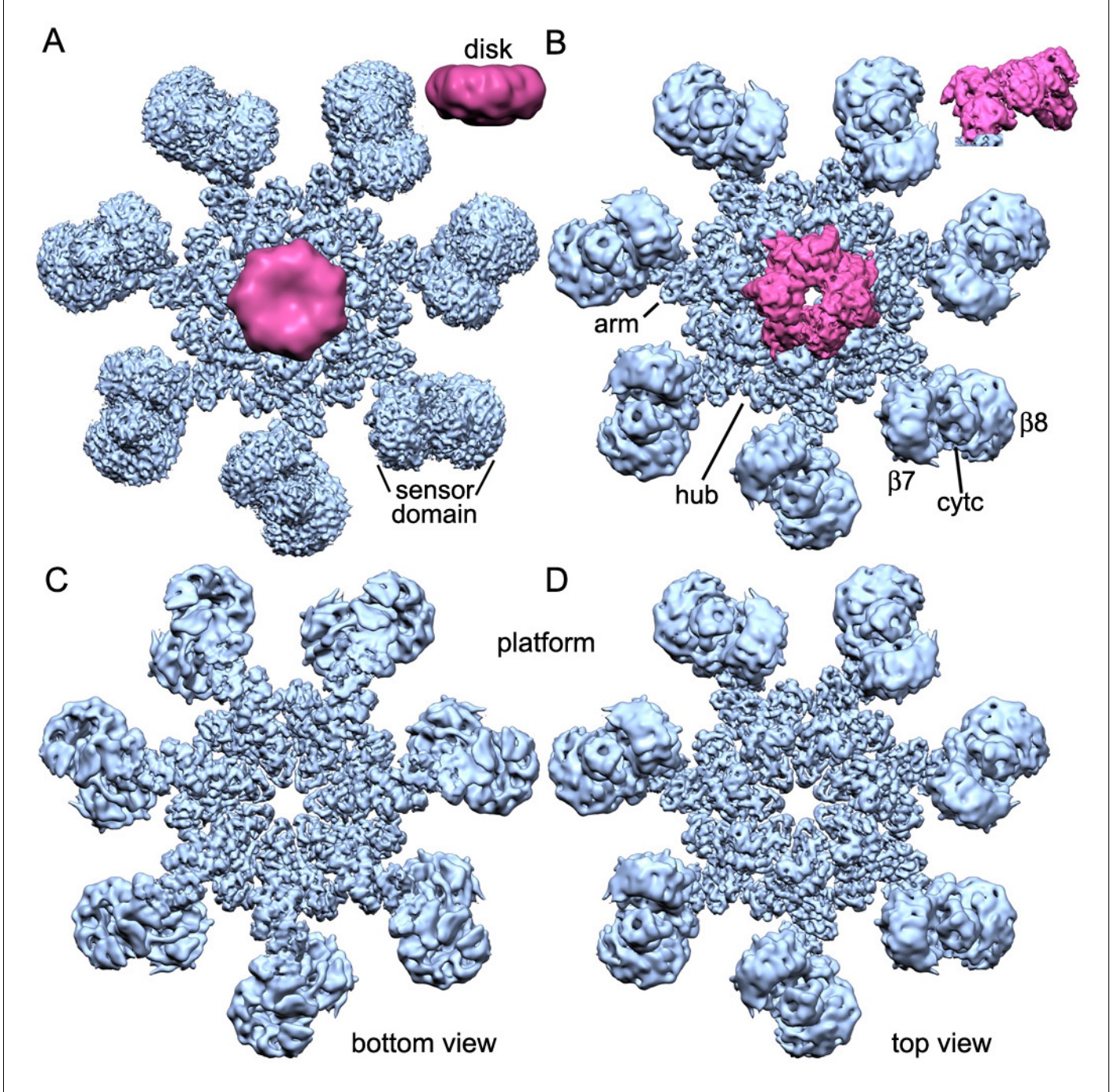

**Figure 1.** Overview of 3D density maps for the active apoptosome. (**A**) A top view of the global 3D density map is shown with the platform in blue and the blurred CARD disk in magenta. **Inset**: The CARD disk was low pass filtered to ~14 Å resolution and is shown from the side. (**B**) Top view of the composite 3D density map. The sensor domain (with β-propellers (β7, β8) and cytochrome c) and the CARD disk are from maps obtained with focused 3D classification at ~6 Å resolution. The strong tilt of the acentric disk is shown in the inset. (**C, D**) Bottom and top views of the platform. See also Figure 1 — table and figure supplement 1.

The following source data and figure supplement are available for figure 1:

**Source data 1.** Data collection and refinement statistics.

**Figure supplement 1.** Sample and structure characterization for the active apoptosome.

## Results

### Structure determination

We assembled an active human apoptosome from Apaf-1 and a two chain pc-9 with cytochrome c and dATP in Buffer A. Procaspase-9 apoptosomes were run on a linear glycerol gradient and visualized by SDS-PAGE to reveal the expected components (*Figure 1—figure supplement 1A*). We also used a fluorescence assay to monitor the LEHDase activity of pc-9 apoptosomes used in this work (*Figure 1—figure supplement 1B*). For single particle work, we froze freshly assembled samples on holey carbon grids and collected super-resolution movies on a Titan-Krios electron cryo-microscope equipped with an energy filter to remove inelastically scattered electrons and provide better contrast for alignments (Materials and methods). After movie frame corrections for global movements during imaging (*Li et al., 2013*), ~135,000 particles were extracted from the resulting micrographs and processed by 2D and 3D classification in RELION (*Scheres, 2012*) to identify the best particles (~93,000). These particles were 'polished' and refined with seven-fold symmetry (c7) in RELION to produce a map with a global resolution of 4.1 Å, as determined with a gold standard FSC (*Figure 1—figure supplement 1C*; *Scheres, 2014*, *2012*). However, the resolution is not isotropic. As estimated by ResMap (*Kucukelbir et al., 2014*) the central hub is at a nominal resolution of 3–4 Å and this is supported by the visibility of larger side chains. The extended arm and peripheral sensor domains are in the range of 4.5–10 Å resolution (*Figure 1—figure supplement 1E*). A sharpened and normalized 3D map from the global 3D refinement was rendered as an iso-surface to show the wheel-like geometry of the apoptosome, which has dimensions of 270 x 270 x ~75 Å. Many rod-like α-helices are visible in the map that dominate the central hub and seven arms, while a cylindrical CARD disk sits atop the platform and shows little structural detail (*Figure 1A* and inset). A previous model of an extended Apaf-1 (*Yuan et al., 2013*; PDB 3J2T) was docked into the map with some rebuilding (Materials and methods) and pertinent statistics are summarized in *Figure 1—source data 1*.

While this work was being completed, a near atomic structure was published of the ground state human apoptosome (*Zhou et al., 2015*; PDB 3JBT), with improved resolution for the V-shaped sensor domain formed by tandem 7- and 8-blade β-propellers. This prompted us to re-analyze this region in our map of the active apoptosome with focused 3D classification in RELION, coupled with local remodeling (*Scheres, 2012*; *Zhou et al., 2015*; Materials and methods). To complete our analysis, we used focused 3D classification to reveal the architecture of the acentric CARD disk at ~5.8 Å (Materials and methods). A model for the disk was constructed by rigid-body docking of individual Apaf-1 and pc-9 CARDs from a crystal structure (*Hu et al., 2014*; PDB 4RHW) into the improved 3D map with Chimera (*Pettersen et al., 2004*). A composite 3D map for the active apoptosome was constructed by zoning with appropriate domain models in global and local maps with Chimera, and these density maps were combined for display (*Figure 1B–D*). Gold standard FSC curves were computed from independent half volumes using the entire global map, and for local maps containing the sensor domain with cytochrome c and the unblurred CARD disk. This provided estimated resolutions of ~4.1, 6.1 and 5.8 Å, respectively. Model versus map FSC curves suggest that over-fitting is not a major issue. (*Figure 1—figure supplement 1C,D*).

An over-view of the model for the heptameric platform is shown with color-coded domains that include the CARD, NBD, helix domain 1 (HD1), winged helix domain (WHD), helix domain 2 (HD2) and β-propellers (*Figure 2A,C*; top views with and without the map). In addition, the disk and a single Apaf-1 molecule are highlighted in the active apoptosome without the map (*Figure 2B*), and a close-up is shown of the disk within the map (*Figure 2D*). Subunit interfaces, the quality of individual Apaf-1 domains and domain boundaries, cytochrome c and the pc-9 CARD are shown in *Figure 2—figure supplements 1–3*, with color coding that is used throughout this work. We also created a sequence alignment in Chimera (*Pettersen et al., 2004*) for Apaf-1, Dark and CED-4 apoptosomes (*Figure 2—figure supplement 4*). This alignment provides detailed information on secondary structure motifs and is a useful guide when discussing this large structure. The heptameric platform appears to be mostly unchanged during pc-9 activation relative to the ground state, with an rmsd of 1.7 Å for Cα's, when comparing Apaf-1 and cytochrome c in the two complexes (*Zhou et al., 2015*; this work). However, the active apoptosome contains an acentric disk formed by Apaf-1 and pc-9 CARDs, with a catalytic domain of pc-9 bound to the central hub in some particles (as discussed later; *Yuan et al., 2010*, *2011a*, *2013*). Importantly, structures are now available for apoptosomes

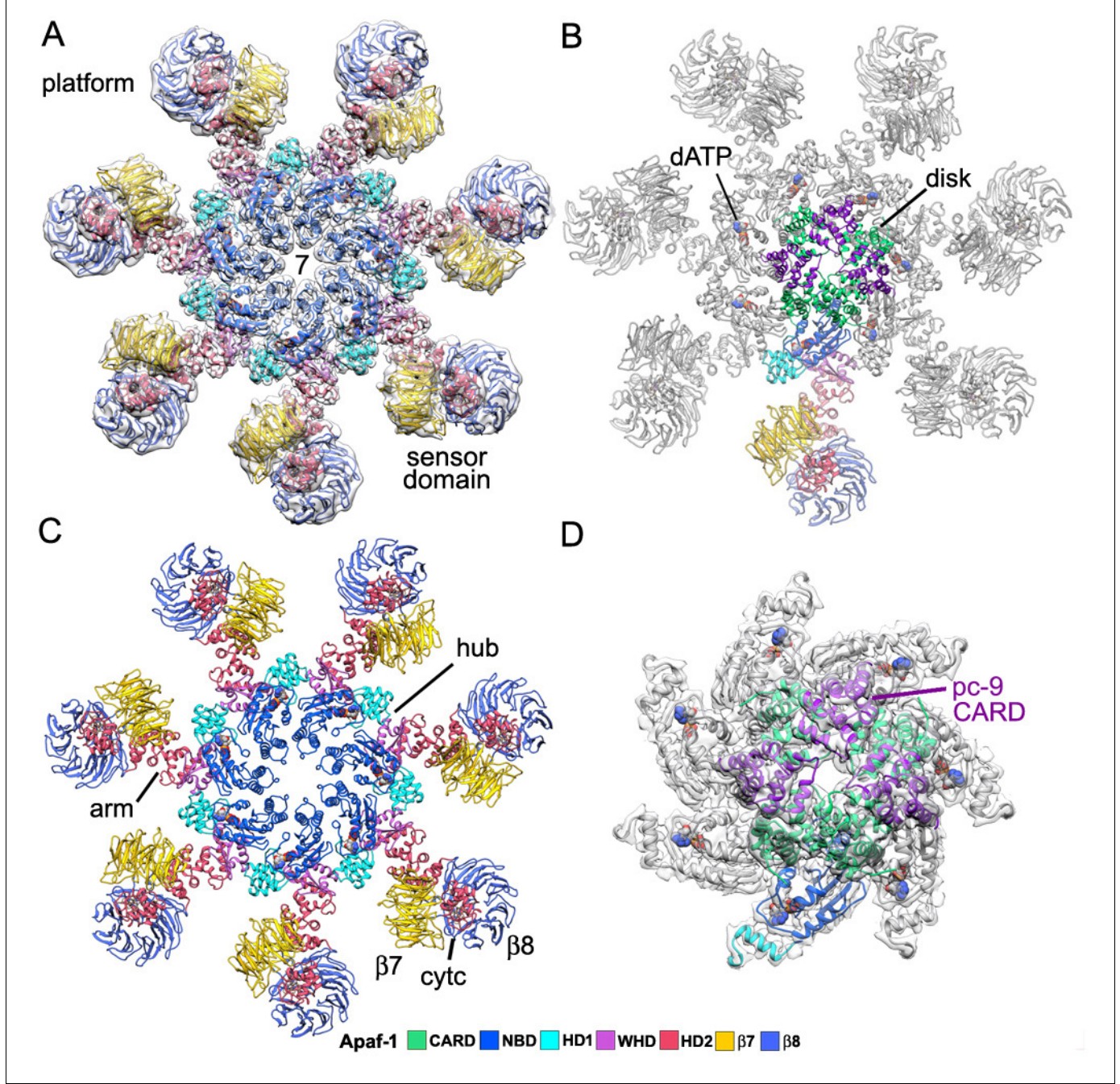

**Figure 2.** Model of the active apoptosome. (**A, C**) Top views are shown of the heptameric platform from the apoptosome model, with and without the composite 3D map. Apaf-1 domains are highlighted in color (see color key). (**B, D**) A top view is shown of the complete model of the active apoptosome without the 3D map. In addition, a close-up of the acentric disk and part of the NBD ring is shown within the composite map (panel **D**). Domains in a single Apaf-1 and the acentric CARD disk are color-coded, relative to the remaining Apaf-1 subunits, which are shown in grey. See also *Figure 2—figure supplements 1–4.*

The following figure supplements are available for figure 2:

**Figure supplement 1.** An Apaf-1 subunit with a bound pc-9 CARD within the active apoptosome.

**Figure supplement 2.** Segmented domain maps for the Apaf-1 subunit with a bound pc-9 CARD.

**Figure supplement 3.** Representative electron density is shown for the four domains of the central hub and HD2 arm.

*Figure 2 continued on next page*

*Figure 2 continued*

**Figure supplement 4.** Sequence alignment for Apaf-1, Dark and CED-4 subunits from their respective apoptosomes, prepared in Chimera (*Pettersen et al., 2004*).

from *C. elegans* (CED-4; *Qi et al., 2010*), *D. melanogaster* (Dark; *Pang et al., 2015*; *Yuan et al., 2011b*; Cheng et al., unpublished) and *H. sapiens* (Apaf-1; *Zhou et al., 2015*; this work), which together provide a wealth of structural data to interpret function.

## Domain and subunit interactions

The central hub of the apoptosome 'wheel' contains seven copies of the NBD, HD1 and WHD in two nested rings (*Figure 3A*). In addition, HD2 forms an arm in each subunit that extends outwards from the hub and ends in a V-shaped sensor domain formed by 7- and 8-blade β-propellers with cytochrome c bound between them (*Figure 2* and *Figure 2—figure supplements 1*, *2*). The central hub is formed by lateral interactions between adjacent NBDs including a ring of paired α-helices that surrounds the central pore (*Figure 3B,C*). This cylindrical picket-fence is formed by helix α12, the initiator specific motif (ISM; *Danot et al., 2009*), which is connected to helix α13 by a long loop that lines the pore itself. The two α-helices in this motif are held together by a hydrophobic interface (*Figure 3D*), while an extensive network of hydrogen bonds and salt bridges mediate interactions between adjacent α12-loop-α13 motifs (*Figure 3E*). Possible interactions involve: Ser213, Glu210' and Gln214'; Arg215, Glu222 and Thr204' and Asn219, Glu221 with Asn201' (where prime denotes the adjacent subunit; not shown). The nucleotide dATP is bound in a pocket at the interface between the NBD and HD1.

An outer ring in the hub is formed by interactions between an HD1 and the WHD in adjacent subunits to compliment extensive interactions between HD1-WHD pairs in each subunit (*Figure 3F*). There are also interactions between the NBD and the HD1-WHD pair within a single subunit. The HD1-WHD loop is involved in dATP binding and interacts with the NBD of an adjacent monomer. This surface is formed in part by possible interactions between Ser356 and Ser358 in the HD1-WHD loop to Asp271 and Ser272 in an extended region, which follows strand β4 in the NBD (*Figure 3G, H*; *Figure 2—figure supplement 4*). However, some of these interactions are in the 4.5–5 Å range and thus, could be mediated in part by water molecules that have not been resolved. Overall, the large number of hydrogen bonds and salt bridges between domains and subunits may be responsible for the tendency of the apoptosome to disassemble at higher salt concentration. Also note that the WHD-HD2 interface is similar to that observed in crystal structures of an inactive Apaf-1 (*Riedl et al., 2005*; *Reubold et al., 2011*). This interface stabilizes the extended HD2 arm and side chain densities are visible for many residues in the WHD and HD2.

## Sensor β-propellers and cytochrome c

Sensor domains in the apoptosome are formed by 15 WD40 repeats that form tandem 7- and 8-blade β-propellers in the C-terminal half of the molecule (*Yuan et al., 2011a*, *2013*; *Reubold et al., 2011*). This region in the global density map is at lower resolution (*Figure 1* and *Figure 1—figure supplement 1*). The radial fall-off in resolution may be due to rotational alignment errors in this thin and extended particle, coupled with local flexibility. However, a focused 3D classification (Materials and methods; *Zhou et al., 2015*) allowed us to obtain an improved density map with a resolution of ~6.1 Å for this region. Structures of human Apaf-1 β-propellers (*Reubold et al., 2011*; *Yuan et al., 2013*; 3J2T) were docked into the density map, along with bovine cytochrome c and a close-up view reveals their interactions (*Figure 4A,B*).

Beta propellers are cylindrically symmetric with each blade formed by a 4-stranded anti-parallel β-sheet (strands a-d). The a-strand of each β-sheet is near the central pore while the d-strand is located at the outer surface of the cylinder. Individual blades of 7- and 8-blade β-propellers are well resolved in the map (*Figure 2—figure supplement 2*). Starting at the distal end of the HD2 arm, a cleft is present at the base of the V-shaped sensor domain which accommodates four helices of HD2 (α29-α32) to form an interface reminiscent of a ball and socket joint, as viewed from front and back

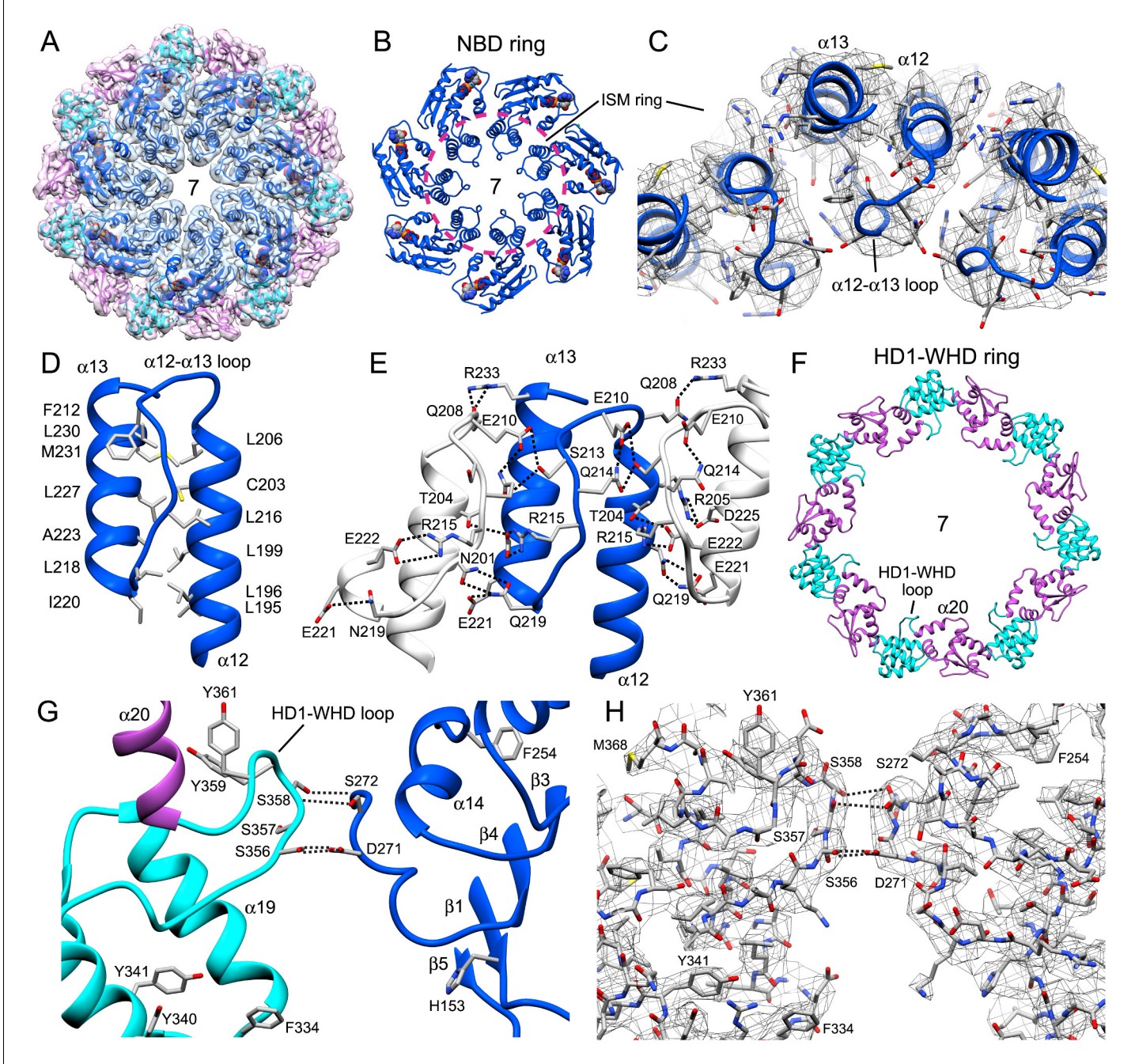

**Figure 3.** Interactions within the central hub. (A) The central NBD ring and the outer HD1-WHD ring are shown as ribbons in a segmented map of the hub. (B, C) The α12-α13 helices and linker between them interact with other copies of this motif to form a cylindrical picket fence that lines the central pore (see pink dashed circle, ISM-ring). These features are shown with atomic interactions in the map. (D, E) The α12-loop-α13 motif is stabilized by an extended hydrophobic core and interacts with adjacent motifs in the ring through numerous hydrogen bonds and salt bridges. (F) An overview is shown of the outer HD1-WHD ring. (G, H) The HD1-WHD loop forms part of the dATP binding pocket and interacts with an extended region that replaces helix α15 in the NBD of an adjacent subunit. Possible interactions are indicated by dashed lines.

(*Figure 4B,C* and inset). The socket is formed by loops between blades 6 and 7 (Ser1169-Gly1178) and between blades 7 and 8 in the 8-blade β-propeller, along with the outer surface of blades 7 and 1 in the 7-blade β-propeller. In the HD2 arm, helices α30 and α31 and the loop between them are inserted into this interface, while helix α29 makes contact with blade 1 of the 7-blade β-propeller.

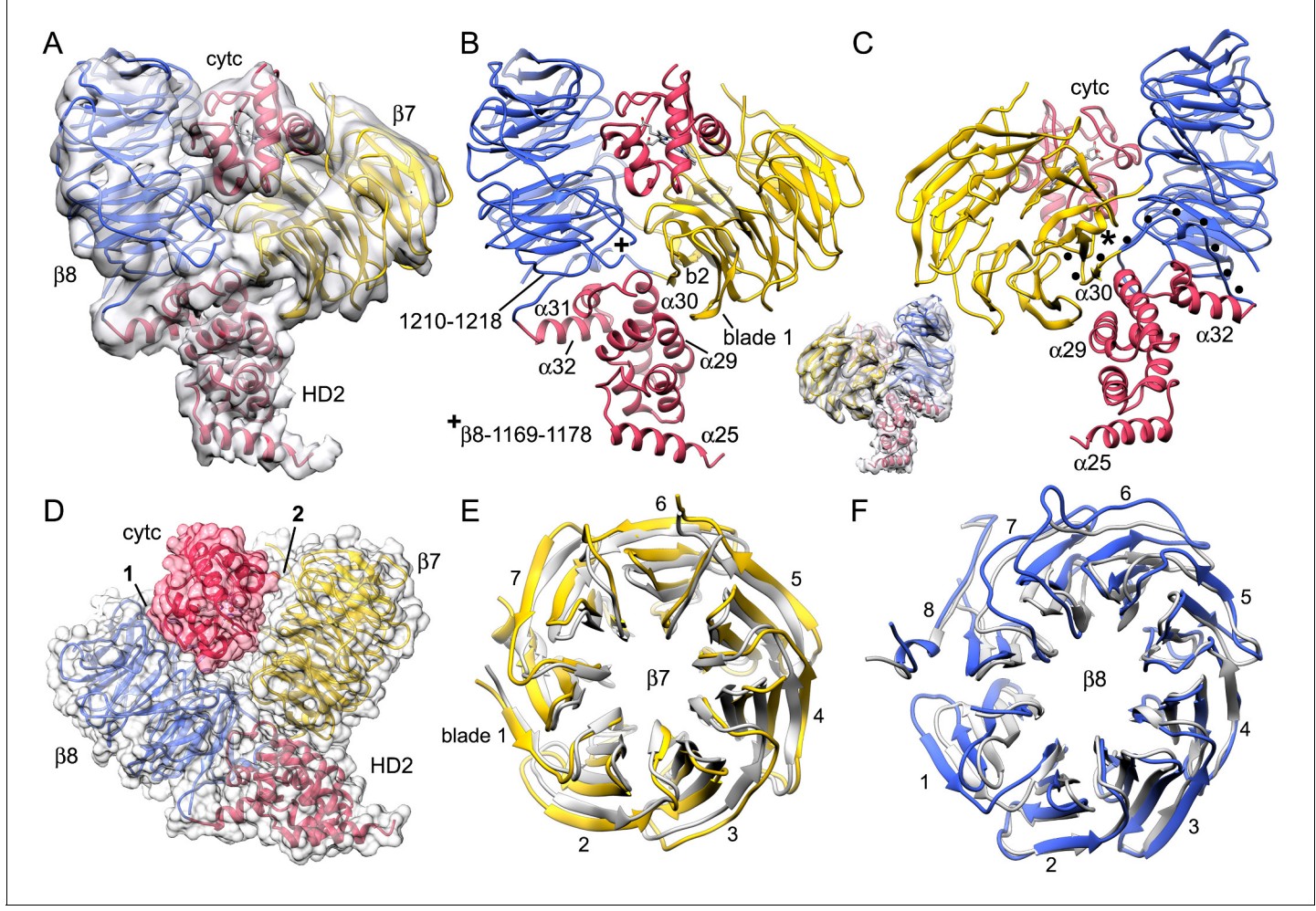

**Figure 4.** The sensor domain and interactions of β-propellers with cytochrome c. (A, B) An overview of β-propellers, cytochrome c and HD2 arm is shown with and without the density map. This view is from the central hub. (C) A reverse view is shown of the sensor domain. The linker from HD2 to the 7-blade β-propeller is indicated with black dots. (D) A calculated molecular surface is shown for the sensor domain super-imposed on color coded ribbon models. Interfaces between cytochrome c and the β-propellers are marked 1 and 2. (E, F) The β-propellers are viewed along their pseudo-symmetry axes from within the V-shaped cleft. The β-propellers are overlayed with their counterparts (in grey) from the crystal structure of mouse Apaf-1. See also *Figure 4—figure supplement 1*.

The following figure supplement is available for figure 4:

**Figure supplement 1.** Connections between cytochrome c and the β-propellers.

The HD2 arm is connected to the 7-blade β-propeller by an unusual linker (*Yuan et al., 2010*) that is present in an Apaf-1 crystal structure (*Reubold et al., 2011*). The HD2-β7 linker starts at helix α32 of the HD2 arm (*Figure 4C*, traced by black dots) and runs along the C-terminal blade of the 8-blade β-propeller, where it replaces the outermost d-strand. The polypeptide chain then crosses over to the 7-blade β-propeller to form the outer d-strand of the last blade (Asp589-Thr615) and then starts the innermost a-strand of blade 1. The interaction between blades 7 and 1 closes the cylindrical fold of the 7-blade β-propeller. Clear density is present for both of the linkers between β-propellers, including a short turn of α-helix in the linker between the 7- and 8-blade β-propellers (see asterisk, *Figure 4C* and *Figure 2—figure supplement 2*).

Cytochrome c is released from mitochondria to trigger apoptosome assembly from Apaf-1 monomers in the cytoplasm and is located between the two β-propellers (*Yuan et al., 2010*, *2013*; *Zhou et al., 2015*). We were able to accurately dock cytochrome c into the improved density

map (*Figure 2—figure supplement 2*). All α-helices in cytochrome c are resolved when the map is viewed at a much higher threshold (not shown) and the overall fit is similar in ground state and active apoptosomes (this work, *Zhou et al., 2015*). Importantly, a rotation of ~90 degrees was required to position cytochrome c, relative to our previous model, due to an ambiguity in the two top docking solutions when fitting with a map at a global resolution of 9.5–10 Å (*Yuan et al., 2013*).

The cytochrome c molecule appears to interact preferentially with the 8-blade β-propeller through an extensive contact region, which is apparent when the map is viewed at a lower threshold, which provides a better estimate of the all atom volume (boxed region 1, *Figure 4—figure supplement 1A*). Additional contacts with the 7-blade β-propeller are found at the bottom of the V-shaped cleft and on the surface opposite from the 8-blade β-propeller (see asterisk and box 2 in *Figure 4—figure supplement 1B–D*). However, contacts to the 8-blade β-propeller are more extensive and involve numerous van der Waals interactions, while a gap of ~3–5 Å is present over much of the interface between cytochrome c and the 7-blade β-propeller (*Figure 4D*, interfaces labeled 1 and 2, respectively). Some possible hydrogen bond and salt-bridge interactions are shown in box 2, which span the gap between cytochrome c and the 7-blade β-propeller (*Figure 4—figure supplement 1D*). In addition, a strong density at the base of the V-shaped cleft may arise from a tryptophan (Trp844) in the 7-blade β-propeller. Two lysine residues from cytochrome c (Lys72 and Lys79) are also present in this region. In total, 7 lysines from cytochrome c are found in the major contact regions and include: lysines 25, 27, 39, 55, 72, 73 and 79 (also see *Yu et al., 2001*). However, higher resolution will be needed to pin down the precise nature of the interactions (also see mutation studies in *Zhou et al., 2015*).

Cytochrome c binding has a direct impact on the structure of the β-propellers, relative to their conformation in a crystal structure of unliganded Apaf-1 (*Reubold et al., 2011*). An overlay of 7-blade β-propellers without and with bound cytochrome c shows a symmetric radial displacement with an rmsd of 1.9 Å (*Figure 4E*). Extensive interactions of cytochrome c with the 8-blade β-propeller may be responsible in part, for a distortion of blades 6, 7, 8 and 1, relative to the crystal structure, while blades 2–5 overlay quite well (overall rmsd of 2.9 Å. *Figure 4F*). This distortion of the 8-blade β-propeller may also arise from the large displacement of the 7-blade β-propeller upon cytochrome c binding, which affects the interface between the two β-propellers, and could reflect altered interactions with helices α29-α32 in HD2.

## dATP binding by Apaf-1 in the assembly path

The dATP binding site in Apaf-1 is located at the NBD-HD1 interface and is formed in part by the Walker A loop and the HD1-WHD loop (*Figure 5A*). Side chain resolution in the central hub allowed us to identify molecular interactions as shown by a cross-section from the density map of the dATP binding site (*Figure 5B*; for clarity only a few hydrogen bonds are shown). In particular, the adenine base is bound in a deep hydrophobic pocket lined by NBD residues Pro120, Pro123, Phe126, Val127, Val162, Leu163, Leu286, Lys290, Ileu294 along with Pro321 from HD1 (*Figure 5C*). The adenine base is in an *anti* configuration relative to the deoxyribose ring and makes hydrogen bonds to the backbone carbonyl and amide of Val127 to help position the fused rings (*Figure 5D*). A conserved arginine (Arg129) is located above the adenine base at the 'top' of the pocket and plays a structural role. We note that the *anti* orientation of the adenine base is present in crystal structures of Apaf-1 with bound ADP (*Riedl et al., 2005*; *Reubold et al., 2011*) and in near atomic structures of Dark and CED-4 apoptosomes (Cheng et al., unpublished; *Qi et al., 2010*). However, the adenine ring in the ground state apoptosome was modeled in a *syn* conformer with the base oriented above the deoxyribose ring (*Zhou et al., 2015*; PDB 3JBT). The *syn* orientation of the adenine base is a higher energy conformer. Therefore, we re-fit the Apaf-1 subunit (PDB 3JBT) into the published density map for the ground state apoptosome using the *anti* conformer of dATP without a Mg$^{+2}$ ion. The resulting structure for dATP is in good agreement with the model of an active apoptosome (rmsd 0.54 Å; *Figure 5—figure supplement 1*).

In our model, the HD1-WHD loop forms the bottom of the nucleotide binding pocket. Both tyrosine side chains in the loop are in good density with Tyr359 forming a hydrogen bond to the γ-phosphate (*Figure 5B,D*). Thus, Tyr359 in the HD1 may act as a sensor for dATP akin to sensor I (Arg265) in the NBD. In addition, the oxygen in the deoxyribose ring may form a hydrogen bond to the amide of Leu322. Importantly, the side chain of Ser325 in helix α18 is located within ~4 Å of the 2' carbon of the deoxyribose ring, and is held in position by hydrogen bonds with the 3' OH in the sugar ring

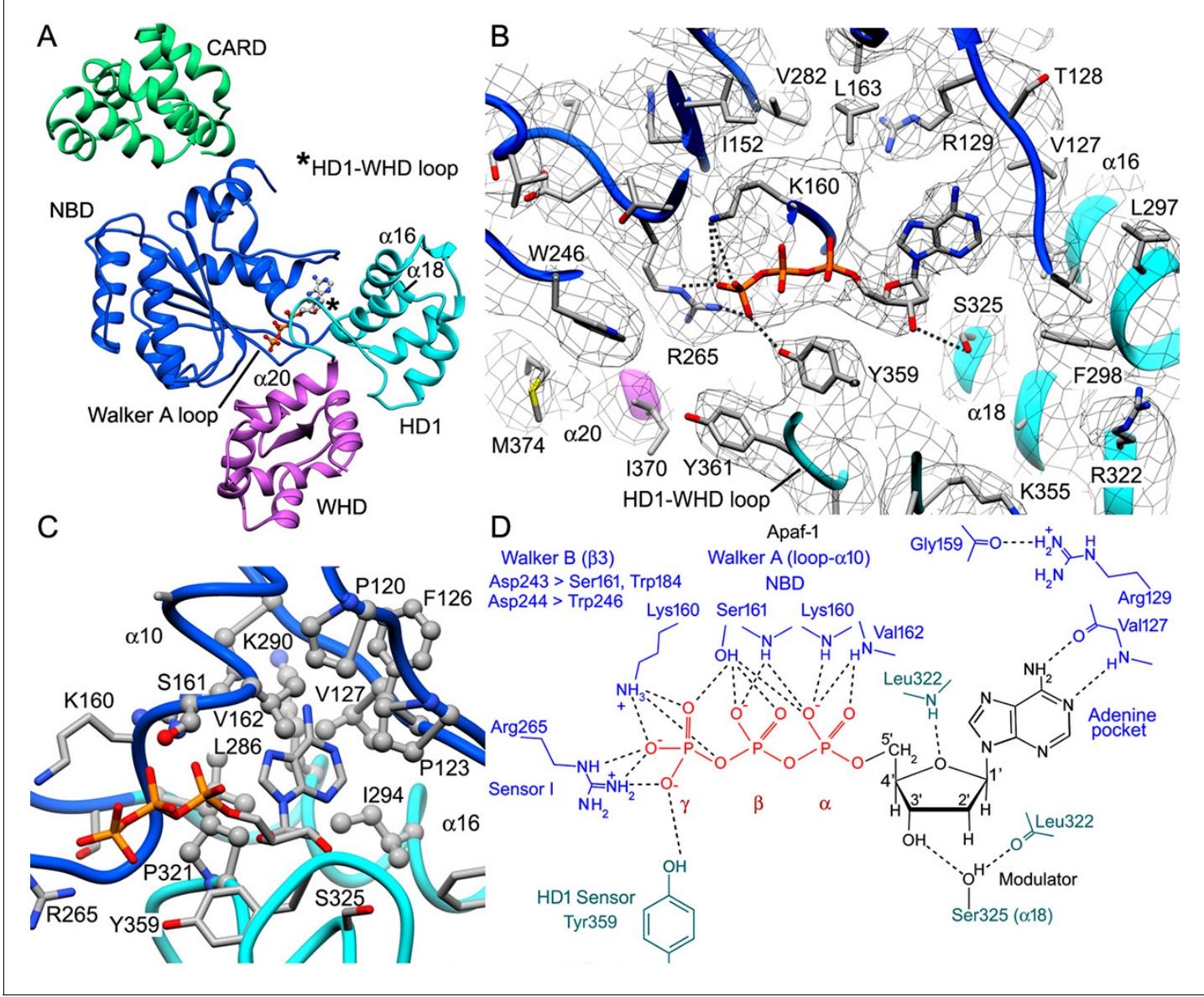

**Figure 5.** dATP binding by Apaf-1 in the apoptosome. (**A**) The dATP binding pocket lies between the NBD and HD1 and is formed in part by the Walker A loop and HD1-WHD loop. dATP is shown in ball and stick representation. (**B**) dATP is shown within the binding pocket overlayed with the density map. A few hydrogen bond and salt bridge indications are indicated by dashed lines. The HD1-WHD loop forms the bottom of the dATP binding pocket. (**C**) The binding pocket for the adenine base is formed mainly by hydrophobic residues, whose side chains are shown in ball and stick representation. (**D**) Interactions of dATP within the binding pocket are summarized in a schematic (see text for details). See also *Figure 5—figure supplements 1, 2*.

The following figure supplements are available for figure 5:

**Figure supplement 1.** The *anti* conformer of dATP and the Apaf-1 subunit (PDB 3JBT; *Zhou et al., 2015*) were fit into the published apoptosome density map (EMD-6480) without a Mg$^{+2}$ ion using real-space refinement with Phenix.

**Figure supplement 2.** A comparison of nucleotide binding pockets.

and the backbone carbonyl of Leu322 (*Figure 5B,D*). In addition, Ser325 may interact with the 2′ OH of the ribose ring when ATP is bound instead of dATP, as found in the crystal structure of Apaf-1 with ADP (*Riedl et al., 2005*; *Reubold et al., 2011*; *Figure 5—figure supplement 2A*). The close

apposition of Ser325 to the 2' carbon of the deoxyribose ring may be responsible for the higher affinity exhibited by Apaf-1 for dATP over ATP (*Jiang and Wang, 2000*). This effect is more obvious at lower nucleotide triphosphate concentrations (*Reubold et al., 2009*). Remarkably, a similar interaction occurs in the Dark apoptosome where a close approach of the Ser325 side chain to the 2' carbon of the sugar ring may be responsible for the absolute preference for dATP during nucleotide-dependent assembly (Cheng et al., unpublished; *Yu et al., 2005*). Thus, Ser325 may act as a modulator that defines the relative affinity for dATP and ATP during assembly of Apaf-1 and Dark apoptosomes.

Residues in the Walker A box (Gly157-Val162; *Figure 2—figure supplement 4*) make numerous hydrogen bond interactions with the triphosphate moiety and are present in a loop that precedes helix α10 and also form the base of this helix (*Figure 5A,D*). In particular, Lys160, Ser161 and Val162 interact with α and β phosphates. The side chain of the absolutely conserved Lys160 interacts with the γ phosphate via three hydrogen bonds. In addition, the side chain of Arg265 (sensor I) makes three potential hydrogen bonds to the terminal phosphate and is stabilized in this position by interactions with Glu366 in the WHD. In the Walker B motif, carboxyl side chains of Asp243 and Asp244 are hydrogen bonded to Trp184 and Trp246, respectively. In addition, Aspartate 243 makes a hydrogen bond to Ser161 in the Walker A motif, and the side chain of Ser161 then interacts with the β phosphate. Aspartate 244 does not interact directly with the γ-phosphate of dATP but is positioned within 5 Å. Hence, water molecules that are not detected at this resolution could mediate a possible interaction.

To investigate the role of nucleotide exchange, we aligned models of closed and extended Apaf-1 on the NBD (*Figure 5—figure supplement 2A*). In the closed state, ADP is bound by residues in Walker motifs in a manner similar to the extended state with bound dATP, except that Arg265 (sensor I) is too far removed from the β-phosphate of ADP to make a salt bridge interaction. However, a major rotation of the NBD-HD1 pair occurs when dATP is bound, such that the HD1-WHD loop rotates into position at the bottom of the nucleotide pocket. This is coupled with a large rotation of the NBD-HD1 pair about the WHD (*Yuan et al., 2013*; *Figure 6*), which positions the NBD, HD1 and WHD to form lateral contacts in the inner and outer rings of the central hub (*Yuan et al., 2010, 2013*). This large rotation also breaks a hydrogen bond between His348 in helix α24 of the WHD and the β-phosphate, which may help stabilize the closed configuration (*Danot et al., 2009*; *Riedl et al., 2005*; *Reubold et al., 2011*; *Figure 5—figure supplement 2A*). Hence, the HD1-WHD loop effectively replaces helix α24 in the transition from a closed to an extended conformation. In addition, a carboxyl triad in the closed state comprises side chains from Asp244 in the Walker B motif, along with Asp392 and Asp439 in the WHD. This side chain grouping should be strongly repulsive but is partially neutralized by Arg265 (sensor I; *Reubold et al., 2011*). As a result of conformational rearrangements during nucleotide exchange, Asp244 in the Walker B motif becomes hydrogen bonded to Trp246, Asp439 in the WHD forms hydrogen bonds with Lys318 (HD1) and Lys391 (WHD), while Asp392 remains in close proximity to Asp439. As predicted, Arg265 does not have to move much in order to interact with the γ-phosphate of dATP during nucleotide exchange and thus, is unlikely to play a direct role in triggering the conformational change (*Figure 5—figure supplement 2A*; *Danot et al., 2009*).

The Apaf-1 monomer has very limited ATPase activity (~4–5 ATPs/Apaf-1/hr; *Reubold et al., 2009*). This is in stark contrast to AAA+ ATPase family members that function as processive machines and catalyze multiple cycles of ATP hydrolysis. Instead, nucleotide exchange may trigger conformational changes in Apaf-1 that allow the platform to assemble. Experimentally, it has been shown that nucleotide hydrolysis does not occur in the apoptosome (*Reubold et al., 2009*; *Kim et al., 2005*). In support of this data, no arginine finger is contributed by an adjacent monomer in the apoptosome to promote catalysis. Furthermore, a bound $Mg^{+2}$ ion also plays an important role in catalysis by AAA+ ATPases. By comparison with the crystal structure of the CED-4 apoptosome, Ser161 and Asp243 in the Walker A motif are predicted ligands for a possible $Mg^{+2}$ ion in the nucleotide binding site of Apaf-1 but a third ligand, equivalent to Lys191 in CED-4, is replaced by Ser186 in Apaf-1, and this side chain is not able to interact with a cation (*Qi et al., 2010*; *Figure 5—figure supplement 2B*). Importantly, no clear density is present in our map of the human apoptosome for a $Mg^{+2}$ ion and modeling with this ion in a previous structure appears to have distorted the model for dATP (*Zhou et al., 2015*). Also note that Buffer A in the frozen pc-9 apoptosome sample contained 1.5 mM $MgCl_2$, 1. 0 mM EDTA and 1.0 mM EGTA. Hence, some free $Mg^{+2}$ ion would

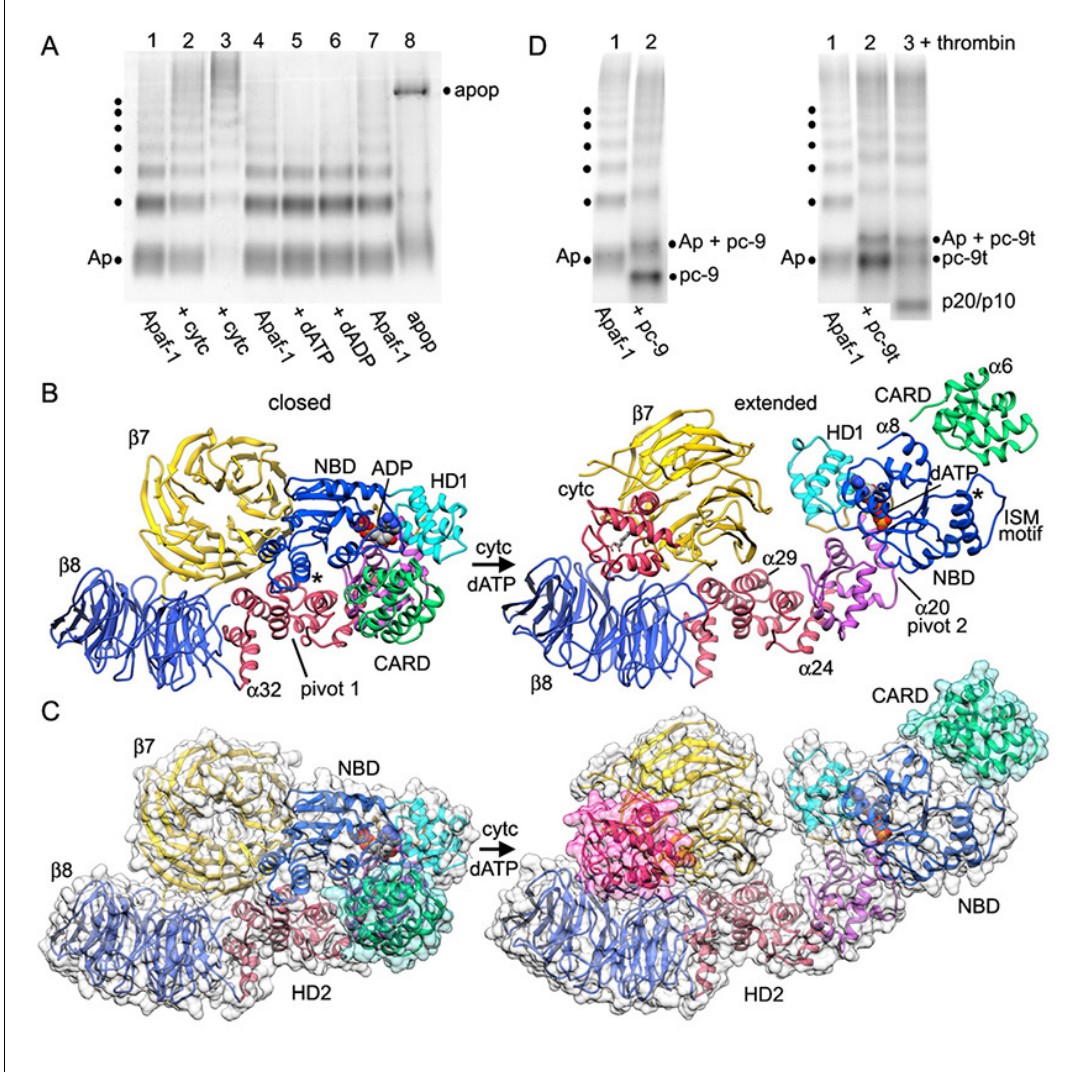

**Figure 6.** Apoptosome assembly. (A) Band shift native gels: (lanes 1–4) interactions of cytochrome c with Apaf-1 create a shifted ladder and higher order aggregation (lane 3) in the absence of dATP; (lanes 4–7) dATP and dADP addition to Apaf-1 do not induce band shifts. The apoptosome is assembled when cytochrome c and dATP are both present (lane 8). (B) Ribbons cartoon showing the closed to extended transition for an Apaf-1 monomer triggered by cytochrome c and dATP binding during assembly. (C) Apaf-1 domains are shown enclosed within calculated surfaces for closed and extended conformations. Reorganization of the NOD creates an HD1-NBD-WHD triad that promotes lateral dimer and ring formation, while the CARD is repositioned to bind pc-9. (D) Band shift native gels demonstrate the accessibility of the Apaf-1 CARD to pc-9. Left panel (lanes 1,2): wild type pc-9 induces a band shift relative to Apaf-1 alone. Right panel (lanes 1–3): a pc-9 mutant with a thrombin site in the CARD-NBD linker also induces a band shift for Apaf-1 and thrombin cleavage releases the catalytic domain (p20/p10) to shift the bands backwards. However, the pc-9 CARD/Apaf-1 bands are still shifted relative to Apaf-1. See also *Figure 6—figure supplement 1*.

The following figure supplement is available for figure 6:

**Figure supplement 1.** The V-shaped sensor domain undergoes a large conformational change upon cytochrome c binding.

have been available for binding to Apaf-1. Intriguingly, a bound Mg$^{+2}$ ion is present in chain A of CED-4 and causes a kink in the triphosphate of ATP. However, Tyr369 in the HD1-WHD loop of CED-4 still makes hydrogen bonds to the γ-phosphate and thus, may act as sensor for HD1. In addition, Thr367 in this loop is hydrogen bonded to both the 3' and 2' OH groups on the ribose ring, thereby subsuming the role of Ser325 on helix α18 of Apaf-1, as this serine has been replaced by a methionine in CED-4 (*Figure 5—figure supplement 2B*).

## Apoptosome assembly

The transition from a closed to an extended conformation of Apaf-1 may allow lateral dimers to form. The sequential addition of further subunits would then overcome entropic barriers to favor apoptosome assembly (*Yuan et al., 2010*; reviewed in *Yuan and Akey, 2013*). In theory, the order of cytochrome c binding and nucleotide exchange may not be fixed during the transition to an extended Apaf-1 monomer (*Reubold et al., 2009*). However, ADP is deeply buried in the binding pocket of the compact monomer in crystal structures (*Riedl et al., 2005*; *Reubold et al., 2011*), which suggests that a major conformational change may be required to facilitate nucleotide exchange. To understand this process, we investigated Apaf-1 assembly with native gradient gels, which are sensitive to changes in mass, shape and charge. However, these experiments were complicated by the presence of a ladder with up to seven bands in purified and functional Apaf-1 preparations. This ladder may be due to lateral subunit interactions at the non-physiological protein concentration. Concerted band shifts were visible when Apaf-1 was incubated with increasing amounts of cytochrome c (*Figure 6A*, lanes 2–3), coupled with a depletion of the original Apaf-1 bands. In addition, higher order bands were formed relative to control lanes with Apaf-1 alone (*Figure 6A*, lanes 1, 4). This suggests that cytochrome c binding may enhance the ability of Apaf-1 to sample more open conformations and in the absence of nucleotide exchange this may lead to aggregation (*Figure 6A*, lane 3).

We next studied the effects of nucleotide triphosphate on Apaf-1 in the absence of cytochrome c. Thus, we added dATP at 1 mM to Apaf-1 and used dADP as a control. In this case, there was no apparent band shift with either nucleotide and no sign of higher order aggregation (*Figure 6A*, lanes 4–7). When dATP and cytochrome c are both added, the aggregation pathway is by-passed and a high molecular weight band is formed that corresponds to the apoptosome (*Figure 6A*, lane 8). Also note that excess cytochrome c in the assembly reaction also induced upward band shifts of unassembled Apaf-1, while the ladder is greatly diminished in intensity. When taken together, these experiments support a sequential model in which cytochrome c binding is required before nucleotide exchange can occur. This idea is consistent with the nature of the conformational changes observed in the transition from a closed to an extended Apaf-1 conformation (see below).

The structure of the human apoptosome reveals subunit interactions required for correct assembly. To further investigate the first steps in assembly, we created a consensus model for full-length Apaf-1 from partial and overlapping models of human and mouse Apaf-1 (*Riedl et al., 2005*; *Reubold et al., 2011*). We then over-layed Apaf-1 structures with and without bound cytochrome c, by focusing the alignment on the WHD and HD2 arm, as these domains remain relatively unchanged during the transition from a closed to an extended conformation. During activation, cytochrome c most likely binds first to the 8-blade β-propeller since this region is accessible and the resulting interface is quite extensive. At this stage, the 8-blade β-propeller and helices α29 to α32 of the HD2 undergo a small upwards rotation relative to the core of the HD2 arm (helices α25 to α28), to bring this region into proximity with the 7-blade β-propeller (*Figure 6B,C*; *Figure 6—figure supplement 1A–C*, *Video 1*). At the same time, the 7-blade β-propeller undergoes a large local twist and rotation to clamp cytochrome c between the two propellers. This rotation breaks interactions between the 7-blade β-propeller, NBD and HD2 arm. However, the 7-blade β-propeller is not able to complete its rotation without clashing with the NBD (*Yuan et al., 2013*; *Figure 6C*). Thus, initial movements of the 7-blade β-propeller may increase domain flexibility in the NOD, as suggested by cytochrome c binding experiments (*Figure 6A*). This flexibility may facilitate nucleotide exchange as the NBD-HD1 pair rotates into a more extended position, since the ADP molecule would become more accessible (*Riedl et al., 2005*; *Reubold et al., 2011*).

**Apaf-1 monomer**

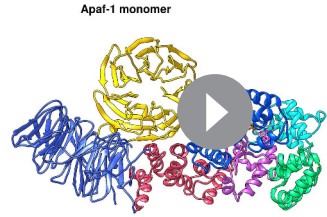

**Video 1.** Conformational change of the Apaf-1 monomer during assembly. This video shows a morph between a model of the closed Apaf-1 molecule and the extended conformation that forms the apoptosome. During this process cytochrome c is bound between the β-propellers and dATP is then exchanged for bound ADP.

As described previously (*Yuan et al., 2013*, *2010*), a large rotation of the NOD occurs about the HD1-WHD interface and the pivot point appears to lie roughly along the axis of helix α20 (*Figure 6—figure supplement 1C* 'pivot 2'; *Video 1*). In this scenario, nucleotide exchange would occur after cytochrome c binds to Apaf-1. This two-step process would stabilize an extended conformation of Apaf-1 to drive apoptosome assembly. Side views of Apaf-1 molecules show the large extension that is achieved (*Figure 6B,C*). This conformational change places domains of the NOD in a position to properly associate with additional Apaf-1 monomers that are in an extended or nearly extended conformation, to form lateral dimers, trimers and higher order oligomers that lead to ring formation (*Figure 6—figure supplement 1D*; *Video 2*).

The consensus model of a closed and inactive Apaf-1 also suggested that the N-terminal CARD may be accessible for interactions with pc-9. This idea is supported by the observation that the Km of Apaf-1 for dATP is lower in the presence of pc-9 and cytochrome c (1.7 to 0.86 uM; *Jiang and Wang, 2000*). To test this idea, we did band shift experiments with native gels in which the Apaf-1 ladder amplified the observed interactions. We added pc-9 or pc-9t (with a thrombin site in the CARD-p20 linker; *Yuan et al., 2011a*) to Apaf-1 and observed a significant upwards shift of all bands in the ladder (*Figure 6D*, left and right hand panels respectively). Thrombin cleavage of the pc-9t linker released the catalytic domains (p20/p10) and the resulting Apaf-1/pc-9 CARD bands were shifted downward on the gel, but were still vertically displaced relative to the corresponding Apaf-1 bands (see lane 1 in *Figure 6D*, right panel). Based on this data, it is clear that the Apaf-1 monomer is able to form a 1:1 complex with pc-9 that is mediated by CARD-CARD interactions. This has implications for the assembly mechanism (see Discussion).

## The CARD disk

Molecular details of the CARD disk have not been visualized in the active apoptosome, due to a possible symmetry mismatch between the disk and central hub (*Yuan et al., 2010*, *2011a*). However, 3D maps calculated with c1 symmetry in EMAN2 revealed additional features including a defined tilt and an acentric location of the disk, relative to the central seven-fold axis (*Yuan et al., 2011a*). To resolve the disk at ~5.8Å resolution we used focused 3D classification without additional refinements in RELION (Materials and methods; *Scheres, 2012*; *Zhou et al., 2015*). In the resulting map, the acentric disk contains seven well-ordered CARDs arranged as a clockwise spiral when viewed from above. At this resolution, CARD-NBD linkers and distinctive features of the 6 helix bundles of Apaf-1 and pc-9 CARDs were resolved, including ~40 α-helices in the disk, which allowed an unambiguous rigid-body docking of the CARDs into the improved map. Moreover, a fourth pc-9 CARD is present at lower occupancy so there are four Apaf-1 CARDs and three or four pc-9 CARDs in the disk at any instant. Apaf-1 and pc-9 CARDs are shown as color coded pairs based on their interactions in the disk along with four CARD-NBD linkers colored in gold (*Figure 7A,B*).

In the disk, four Apaf-1/pc-9 CARD pairs form a quasi-helical spiral and three Apaf-1 CARDs are in direct contact with the NBD ring (*Figure 7C*). The rise of the spiral creates a tilted disk with a large gap between the disk and the NBD ring on one side (*Figure 7A*, top right). An Apaf-1 CARD comes into close contact with the NBD ring and is adjacent to the gap in the clockwise direction, when viewed from above. We defined this Apaf-1 CARD as position 1a in the spiral (*Figure 7B*, in red), while an adjacent pc-9 CARD in the clockwise direction is vertically and laterally offset from the first Apaf-1 CARD and occupies position 2p. Overall, Apaf-1 and pc-9 CARDs are present at odd and even positions around the spiral, respectively, (*Figure 8A,B*). In this notation, Apaf-1/pc-9 CARD pairs are located in positions 1a-2p, 3a-4p, 5a-6p and 7a-8p, where they define

the Apaf-1 platform

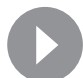

**Video 2.** Structure and assembly of the c7 platform and spiral CARD disk in the human apoptosome. This video documents a model for the stepwise assembly of a c7 platform from Apaf-1 subunits containing the NOD, HD2 arm and β-propellers. The presentation then focuses on the structure and molecular docking of Apaf-1 and pc-9 CARDs within the acentric, quasi-helical disk, which sits atop the platform.

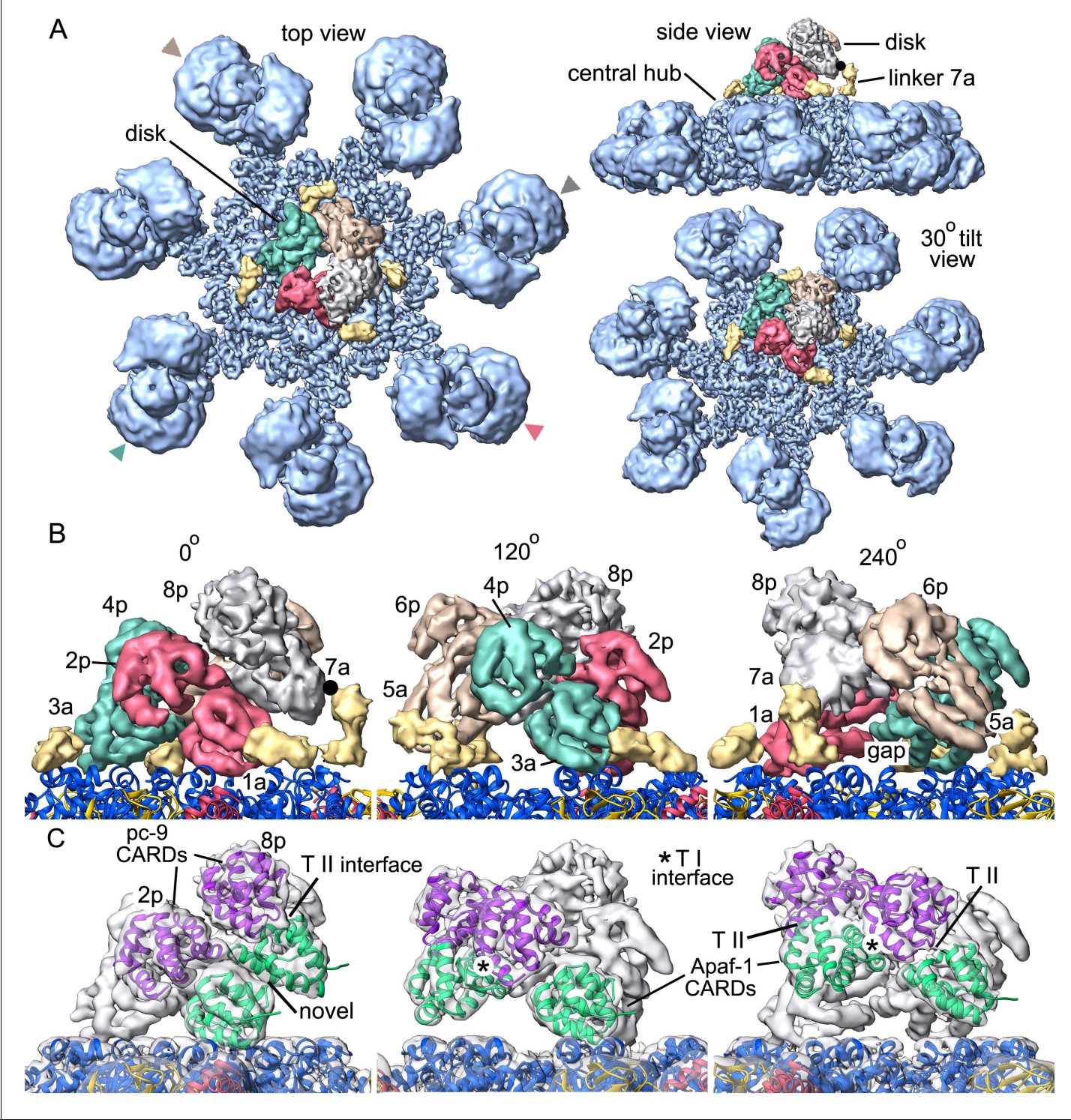

**Figure 7.** Formation of a CARD disk on the active apoptosome. (**A**) A composite map of the active apoptosome reveals density for eight CARDs in the acentric disk shown in top, tilted and side views. Density for the pc-9 CARD in position 8p has been included, although it is present at lower occupancy. Apaf-1/pc-9 CARD pairs that share a Type II interface have been color-coded and Apaf-1 subunits that contribute an ordered CARD to the disk are indicated by color-coded arrow heads. Apaf-1 CARD-NBD linkers are shown in gold and the linker to CARD 7a is marked with a black dot. (**B**) Close-up views are shown of the CARD disk, viewed in three equi-spaced angular orientations about the center of the apoptosome. The α-helical nature of the CARDs is clearly visible along with the progressive tilt of the pairs as they spiral about the center of the disk. CARD pairs are rendered as surfaces and individual CARD positions are indicated (1a, 3a, 5a and 7a for Apaf-1; 2p, 4p, 6p and 8p for pc-9). Individual linkers are shown as segmented density in

*Figure 7 continued on next page*

*Figure 7 continued*

gold for the four Apaf-1 CARDs. (**C**) Similar views to panel B are shown, but as transparent surfaces with docked ribbon CARD models color-coded green for Apaf-1 and purple for pc-9. Only two CARD pairs are shown with ribbons in each panel to highlight their interactions. Some Type I and II interfaces are indicated. CARD-NBD linkers have been omitted for clarity.

a one-start, quasi-helical spiral. An Apaf-1 CARD in position 7a is located above the gap where it contacts the top of the Apaf-1 CARD in position 1a (*Figure 7B*, left panel). Finally, weaker density is present at position 8p in the average map. However, this region could be visualized in a normalized map at a lower threshold. This density may correspond to a pc-9 CARD, which is present at lower occupancy and thus, would interact with the Apaf-1 CARD in position 7a to complete the fourth CARD pair. The pc-9 molecule at position 8p may be readily exchangeable, while the other pc-9 CARDs make more extensive contacts in the disk.

In total, four of the seven Apaf-1 CARDs in the apoptosome are used to construct the disk. In addition, CARD-NBD linker densities connect the C-terminal α-helix of each CARD to helix α8 of its respective NBD, when viewed at an appropriate threshold (in gold, *Figure 7B*). The linkers in this figure are clearly defined after map segmentation with SEGGER (*Pintilie et al., 2010*) and are shown at a lower threshold. Molecular models of the linkers have not been built due to the limited resolution of these features. The four Apaf-1 subunits that contribute CARDs to the disk are marked by color-coded arrow heads (*Figure 7A*). The remaining three Apaf-1 CARDs in the apoptosome are not added to the top of the disk to form a third layer, presumably due to restraints imposed by the length of their CARD-NBD linkers. In addition, steric clashes or increasing distortions in the CARD-CARD interfaces of the quasi-helical spiral (see below) may preclude the addition of a third layer to the disk. We surmise that the three excluded Apaf-1 CARDs are disordered since they are not visible, but they may contribute to the propensity of the particles to aggregate. In addition, band shift experiments with Apaf-1 monomers and pc-9 suggest that excluded Apaf-1 CARDs may, in principle, bind to pc-9 molecules in the absence of steric restraints. However, a measured stoichiometry of 2 to 5 pc-9 molecules per apoptosome under various conditions, places an upper limit on the number of Apaf-1 CARDs that may recruit pc-9 molecules (*Malladi et al., 2009*; *Yuan et al., 2011a*; *Hu et al., 2014*). Thus, Apaf-1 CARDs in the disk and perhaps one excluded CARD may be able to recruit pc-9 molecules to the apoptosome.

The CARD spiral is located acentrically relative to the seven-fold symmetry axis of the apoptosome and rests in a tilted position on the NBD ring (*Figure 7*). In all of the Apaf-1 CARDs, the N-terminal helix is positioned closer to the NBD ring, while the adjacent C-terminal helix is located above the N-terminal helix with both termini facing outwards from the disk. Thus, the C-terminal helix of each Apaf-1 CARD is positioned so that its linker to the NBD is located on the outside of the disk (*Figure 7A,B*). Contacts made by Apaf-1 CARDs to NBDs in the central hub vary due to their position in the spiral. At position 1a, the first 10 residues in the N-terminal α-helix of the Apaf-1 CARD make contacts with helices α8 and α13 in two adjacent NBDs. At position 3a the N-terminal helix of the Apaf-1 CARD may interact with helices α8 and α11 in a single NBD. Due to the gradual rise of the spiral, the Apaf-1 CARD at position 5a makes weaker contacts with helix α11 in a single NBD, while the Apaf-1 CARD at position 7a does not contact the central hub, although it is attached to its NBD by a CARD-NBD linker (*Figure 7B*). A gap is present between the CARD disk and the platform at this position (marked '7a', *Figure 7B*, left panel).

Surprisingly, only three well ordered pc-9 CARDs and a single more weakly bound molecule are present in the disk. An oversized mask was used to compute the disk structure with focused 3D classification; hence no density has been omitted from the final map due to restrictions imposed by the mask. Thus, our data suggests that seven well ordered CARDs form the disk. In addition, no direct contacts are present between pc-9 CARDs and the central hub, and all pc-9 domains are located on the top surface of the disk-like spiral with their N- and C-termini pointing towards the outside. This would allow easy access of the long linker from pc-9 CARDs in the disk to catalytic domains bound to the apoptosome and in bulk solution.

The overall architecture of the disk is distinct from a recent model, which suggested that the disk would contain three layers of alternating Apaf-1, pc-9 and Apaf-1 CARDs (*Hu et al., 2014*). This model was based on a crystal structure of a CARD heterotrimer in which a single pc-9 CARD is

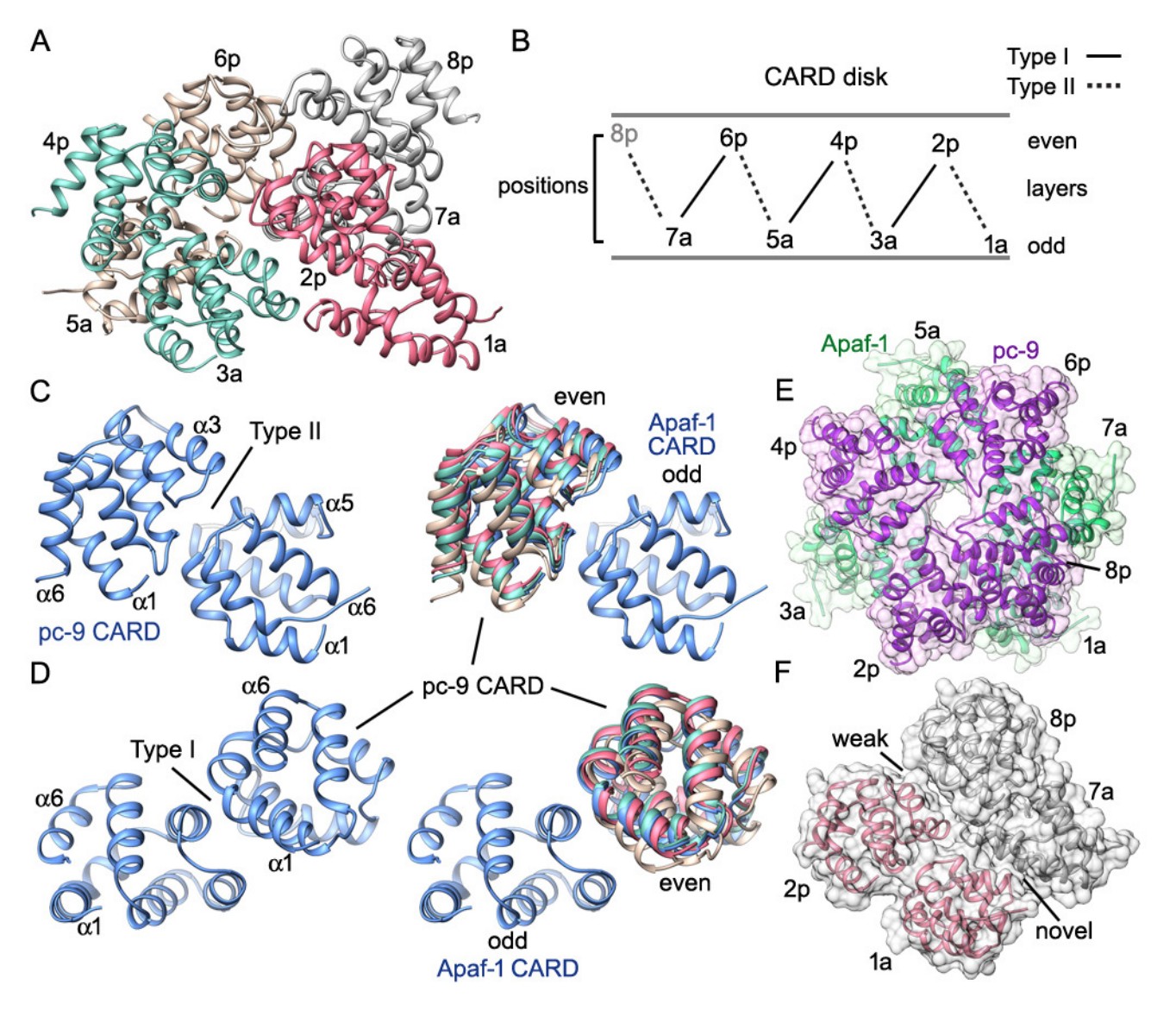

**Figure 8.** CARD packing in the disk. (A) A ribbons diagram of the disk is shown with color coded Apaf-1/pc-9 CARD pairs and individual positions labeled. (B) Schematic of CARD positions in the disk with Type I and II interfaces indicated. (C, D) Alignment of pc-9 CARDs relative to their respective Apaf-1 CARD in Type II and Type I interface pairs. The crystal structure pair (PDB 4RHW) is in blue. (E) Top view of the disk with a calculated surface. Individual CARD positions are indicated. (F) A side view of CARD pairs is shown at the junction between the first and fourth pair, where a dislocation is present in the spiral disk. A weak and quite open interface is present between adjacent pc-9 CARDs (2p, 8p) while adjacent Apaf-1 CARDs (1a, 7a) form a novel interface. See also *Figure 8—figure supplement 1*.

The following figure supplement is available for figure 8:

**Figure supplement 1.** Catalytic domains of pc-9 are parked on the central hub of the active apoptosome.

sandwiched between two Apaf-1 CARDs and used a crystal structure of Death Domain complexes as a template (*Hu et al., 2014*; reviewed in *Ferrao and Wu, 2012*). However, coordinates for this model are not available. The unusual arrangement of Apaf-1 and pc-9 CARDs in the crystallized heterotrimer creates two interfaces, which were denoted Type I and II, with the Type I interface being similar to the contact surface in the structure of an Apaf-1/pc-9 CARD dimer determined previously (*Qin et al., 1999*). The disposition of Type I and II interfaces in the disk is shown in *Figures 7C* and *8B–D* with Type II interfaces occurring between CARDs in color-coded pairs (*Figure 8A,B*). Two

overlays are shown for the relevant CARD pairs centered on Type II and Type I interfaces with color coded pc-9 CARDs, after alignment to the appropriate Apaf-1 CARD in the crystal structure (*Figure 8A,C,D*; *Hu et al., 2014*). In general, the interfaces become more distorted in moving clockwise around the disk from position 1a, due to the spiral geometry of the CARDs. The rmsd for Cα backbones of 'sequential' Apaf-1/pc-9 CARD pairs, aligned with the Apaf-1 CARD of the relevant crystal pair are 3, 3.2 and 5.1 Å for the Type II interface and 2.1, 1.5 and 3.2 Å for the Type I interface. The '7a-8p' CARD pair was not included in this analysis since the crystal structure was used to model this pair. The CARD spiral results in a rather open interface between two adjacent pc-9 CARDs at positions 8p-2p, and a novel interface is formed between Apaf-1 CARDs at positions 7a-1a (*Figure 8E,F*).

Mutations were made in Apaf-1 and pc-9 CARDs using the crystal structure of the heterotrimer as a guide (*Hu et al., 2014*; PDB 4RHW). The zig-zag packing of three consecutive CARDs in the disk at positions 1 to 3, 3 to 5 and 5 to 7, is due to the presence of alternating Type I and II interfaces, as described in the crystal. Hence, mutations that disrupt Type I and Type II interfaces are supported by our model of the disk. These mutations would disrupt CARD-CARD interactions and may decrease pc-9 binding and activity on the apoptosome. Two additional Apaf-1 CARD mutations (K58E, K62E) were found to greatly diminish pc-9 activation and occur outside of interfaces I and II (*Hu et al., 2014*). These mutations are located on the bottom surface of the disk-like spiral and may interfere with Apaf-1 CARD interactions to the NBD ring. These twin point mutations are not positioned where wild type residues may interact with pc-9 catalytic domains and participate in activation. However, these two point mutations suggest that proper disk formation may be a pre-requisite for proper activation of pc-9 zymogens.

## A pc-9 catalytic domain on the hub

Procaspase-9 catalytic domains consist of two subunits denoted p20 and p10 that are generated by zymogen cleavage. We showed previously that density for catalytic domains of a single pc-9 could be identified on the central hub, adjacent to the disk (*Yuan et al., 2011a*, *2010*). With this in mind, we re-processed our pc-9 apoptosome data with c1 symmetry in EMAN2 (*Tang et al., 2007*) and used e2refinemulti.py for 3D classification (Materials and methods). A similar feature for pc-9 was found on the central hub with this large data set, and the density accommodates a single copy of the p20/p10 domains. However, the density was not modeled precisely due to local flexibility of this feature (*Figure 8—figure supplement 1A*). Intriguingly, we could not detect the pc-9 catalytic domain with a focused 3D classification approach in RELION due in part to its small size and local heterogeneity. Refinement with c1 symmetry in EMAN2 uses a density based approach in real space and may reveal the pc-9 catalytic domain at the expense of some blurring of the disk. The success of this approach may also be due to the use of supervised 3D classification, which allowed the identification of particles that may contain bound pc-9 catalytic domains, prior to 3D refinements. We also note that c1 refinement in EMAN2 resolved the platform at α-helical resolution (not shown). We suspect that pc-9 catalytic domains may also bind at sites adjacent to the one identified on the central hub in some particles, as this might explain, in part, why this small (~40 kDa) feature was lost during focused 3D classification in RELION. We also surmise that pc-9 catalytic domains may be bound in a dynamic manner, since only ~50% of the particles contained the density in the aligned location.

We next determined the most likely binding site(s) for the p20/p10 domains of pc-9 in relation to the unblurred disk. Thus, we aligned the composite 3D map with a well resolved disk relative to the map refined with c1 symmetry using Chimera. This alignment was guided by the acentric and tilted disk, since the 7 spokes are similar in both maps. Intriguingly, the catalytic domain can be located in two possible positions that are adjacent to a pc-9 CARD at position 2p within the CARD disk. For simplicity, we have shown the position with the highest cross-correlation, while the next most likely candidate with a similar probability, is rotated by one Apaf-1 subunit in the clockwise direction (*Figure 8—figure supplement 1B,C*). This proximity would allow pc-9 catalytic domains parked on the hub to be connected to CARDs at all positions in the disk due to the long CARD-p20 linker.

## Discussion

In previous studies, cryo-EM and single particle methods were used to determine the structure of human apoptosomes with bound pc-9 CARDs at ~9–10 Å resolution. We then docked domains from

crystal structures into the 3D map to create a model for the apoptosome. This analysis identified plausible conformational changes of Apaf-1 that may drive assembly (*Yuan et al., 2010*, *2013*), although high resolution details were absent. However, density for the CARD disk was blurred due to a symmetry mismatch with the platform. To extend our studies, we have now determined the first near atomic structure of an active human apoptosome using cryo-EM. In total, we modeled 57 domains in the active apoptosome. Models for the sensor domain with cytochrome c and the CARD disk are based on local maps obtained by focused 3D classification at a resolution of ~6Å. The final structure highlights the unusual architecture of this intrinsic cell death machine, in which a spiral-shaped disk is nucleated from the Apaf-1 platform when pc-9 is bound. In addition, most pc-9 catalytic domains are flexibly-tethered to their respective CARDs in the disk, while catalytic domains from one pc-9 may be parked on the central hub in some particles. Our model provides insights into cytochrome c binding, nucleotide exchange, Apaf-1 assembly, CARD disk formation and pc-9 activation. To our knowledge, this is the first time a CARD disk has been visualized clearly in the context of a platform for apical procaspase activation.

## Apaf-1 assembly

In the current assembly model, cytochrome c binding and nucleotide exchange produce a large conformational change in Apaf-1 (*Yuan et al., 2013*; *Pang et al., 2015*; *Reubold et al., 2009*). We suggest that cytochrome c binding is a pre-requisite for nucleotide exchange during assembly. In particular, we observed a significant band shift and aggregation of Apaf-1 on native gels when cytochrome c was added, while similar changes could not be detected when dATP was added in a separate reaction. Conformational changes due to cytochrome c binding may involve a small rotation of the 8-blade β-propeller and the supporting four α-helices of HD2, along with a much larger rotation of the 7-blade β-propeller about the base of the HD2 arm, to form the appropriate contact surface within the V-shaped sensor region for cytochrome c (*Yuan et al., 2013*). During this reaction, the NBD-HD1 pair must begin to rotate away from the 7-blade β-propeller to avoid a clash, while allowing nucleotide exchange. Alternatively, a hinge-like rotation of the NBD-HD1 pair relative to the rest of Apaf-1 may be dynamic. Hence, the NBD-HD1 pair could fluctuate between ADP and ATP/dATP bound conformations until they are fixed in place by the binding of both ligands. Thus, cytochrome c and a bound nucleotide triphosphate would predispose Apaf-1 to adopt an extended conformation that promotes lateral assembly of subunits to form the ring. In the absence of nucleotide exchange, binding by cytochrome c leads to a local conformational change (or alternatively to a change in the dynamics of the closed form) and subsequent aggregation of Apaf-1 molecules.

The low intrinsic ATPase activity of Apaf-1 (~4–5 ATP per Apaf-1/hr) is probably not required for assembly per se (*Reubold et al., 2009*), as exchange of bound ADP/dADP for a nucleotide triphosphate (dATP, ATP or even GTP at 1 mM) is all that is required to promote conformational changes that drive assembly. Near atomic structures of Apaf-1 monomers (*Riedl et al., 2005*; *Reubold et al., 2011*) and the apoptosome (*Zhou et al., 2015*; this work) suggest that the molecule has diverged from the canonical architecture of AAA+ ATPases. This includes the 'loss' of a stable binding site for a $Mg^{+2}$ ion adjacent to the γ-phosphate of dATP/ATP, which greatly diminishes the intrinsic ATPase activity of the monomer. However, as Apaf-1 folds during translation or shortly thereafter, the protein may bind adenine nucleotide diphosphate or adenine nucleotide triphosphate, as both are present in the cytoplasm. If nucleotide diphosphate is bound, then Apaf-1 will be locked into a closed conformation, until cytochrome c is released from mitochondria in response to intrinsic cell death signals. Alternatively, if newly synthesized Apaf-1 binds a nucleotide triphosphate then a limited hydrolysis activity would ensure that the molecule adopts a closed, inactive conformation to prevent aggregation or further assembly into the cell death machine. Moreover, no detectable ATPase activity has been found in apoptosomes (*Reubold et al., 2009*; *Kim et al., 2005*). This may be correlated with the lack of an arginine finger and the absence of a stably-bound $Mg^{+2}$ ion at the nucleotide binding site.

We also find that pc-9 is able to interact with Apaf-1 to form a 1:1 complex and this step is mediated by CARD-CARD interactions. Hence, Apaf-1 and pc-9 may co-assemble to form an active apoptosome. In this case, preformed Apaf-1/pc-9 complexes would bind cytochrome c and undergo nucleotide exchange. Alternatively, Apaf-1 may adopt an extended conformation triggered by cytochrome c and dATP/ATP, and then bind pc-9 as it interacts with other copies of itself during assembly. Remarkably, the concentration of pc-9 and Apaf-1 in HeLa cervical cancer cells is estimated to

be ~30 nM and ~340 nM, respectively (*Würstle and Rehm, 2014*). This translates to roughly one pc-9 molecule for every 10–11 Apaf-1 molecules and with an assembly efficiency of ~70% this would give a pc-9 to Apaf-1 ratio of roughly 1:7 or 1:8 in active complexes. However, in non-cancer cells the ratio of pc-9 to Apaf-1 may be higher. Even so, this raises the possibility that preformed Apaf-1/pc-9 heterodimers may undergo preferential recruitment to apoptosomes during their assembly, due to the additional cooperativity of disk formation.

## The apoptosome and pc-9 activation

We have presented a structure for the central hub and extended arms of an active apoptosome at near atomic resolution (~3.5–4 Å). Improved structural clarity has revealed the sensor domain and for the first time, we have visualized the packing of CARDs within the acentric disk (*Figure 9*, side and close-up views; *Video 2*). Members of the 6-helix Death Domain family (CARDs and Death Domains) have been tailored to assemble into quasi-helical arrays (reviewed in *Ferrao and Wu, 2012*), which form the heart of various activation machines. These include the human apoptosome (*Yuan et al., 2011a*; *Hu et al., 2014*), the NLRC4 inflammasome (*Zhang et al., 2015*; *Hu et al., 2015*; *Diebolder et al., 2015*), the PIDDosome (*Park et al., 2007*), the Myddosome (*Lin et al., 2010*) and the FAS-FADD death disc (*Wang et al., 2010*). However, the disk in the human apoptosome uses 7–8 CARDs to form a spiral, instead of 10–14 domains found in other Death Domain complexes. Various attempts have been made to reconstitute a disk from individual CARDs of Apaf-1 and pc-9. Recently, a meta-stable complex was observed by gel filtration and estimated to contain 7–8 CARDs. These studies led to the crystallization and structure determination of a unique CARD heterotrimer (*Hu et al., 2014*). In our study, we find that the path of paired Apaf-1/pc-9 CARDs in the disk describes roughly one turn of a spiral. In addition, Type I and II interfaces in the crystal structure of the Apaf-1/pc-9/Apaf-1 CARD trimer (*Hu et al., 2014*; 4RHW) are present in the disk, but with some distortions to accommodate the spiral geometry. Strikingly, only three well ordered pc-9 molecules are recruited to the disk, while a fourth pc-9 is present at lower occupancy.

Procaspase-9 exists as a monomer in solution unlike most caspases which are constitutive dimers (reviewed in *Riedl and Shi, 2004*). However, pc-9 forms an unusual dimer when crystallized without the CARD, in which one substrate binding site is occluded and inactive, while a second site is in an active configuration and binds a peptide inhibitor (*Renatus et al., 2001*). Close inspection of the pc-9 structure suggests that crystal contacts do not play a role in this asymmetry. Recent data have been interpreted to suggest a possible model for pc-9 activation that involves dimerization of p20-p10 catalytic domains (*Boatright et al., 2003*; *Bratton and Salvesen 2010*; *Pop et al., 2006*; *Renatus et al., 2001*), which are attached to the apoptosome by flexible CARD-p20 linkers. Alternatively, tethered pc-9 catalytic domains may bind directly to the platform and/or the disk and be activated (*Yuan et al., 2011a*; *Hu et al., 2014*; *Yin et al., 2006*). Recent computational modeling favors an allosteric mechanism wherein pc-9 catalytic domains bind to the apoptosome, but could not differentiate between monomers or dimers as the active species (*Würstle and Rehm, 2014*). However, recent experiments have shown that chimeric Apaf-1 CARD/ClpP or Apaf-1 CARD/GroEl complexes, which are thought to form ClpP and GroEl heptamers, respectively, are able to activate pc-9 to a similar extent as the Apaf-1 apoptosome using a fluorescence-based proteolysis assay and by monitoring procaspase-3 cleavage (*Hu et al., 2014*). We have verified this remarkable observation for pc-9 added to chimeric ClpP molecules with N-terminal Apaf-1 CARDs (not shown). When taken together, this data would argue against an activation mechanism that requires binding of pc-9 catalytic domains to the central hub formed by Apaf-1 molecules, since this region does not exist in chimeric complexes. These data also suggest that efficient procaspase-3 cleavage can proceed in chimeric complexes in the absence of the Apaf-1 platform. However, these studies do not rule out the possibility that procaspase-3 may interact with Apaf-1 at some point either during or after its activation (*Yuan et al., 2011a* and references therein).

Our structure of an active apoptosome provides additional insights into the mechanism of pc-9 activation. One constraint is provided by stoichiometry measurements, which indicate that a range of 2 to 5 pc-9 molecules may be bound to the apoptosome (*Malladi et al., 2009*; *Yuan et al., 2011a*; *Hu et al., 2014*). In line with this data, our structure suggests that 3 or 4 pc-9 molecules may be tethered to the disk through CARD-CARD interactions, with the caveat that a pc-9 molecule might also interact with Apaf-1 CARDs that are excluded from the disk. We also identified density in some particles for catalytic domains from a single pc-9 that are 'parked' on the NBD ring of the

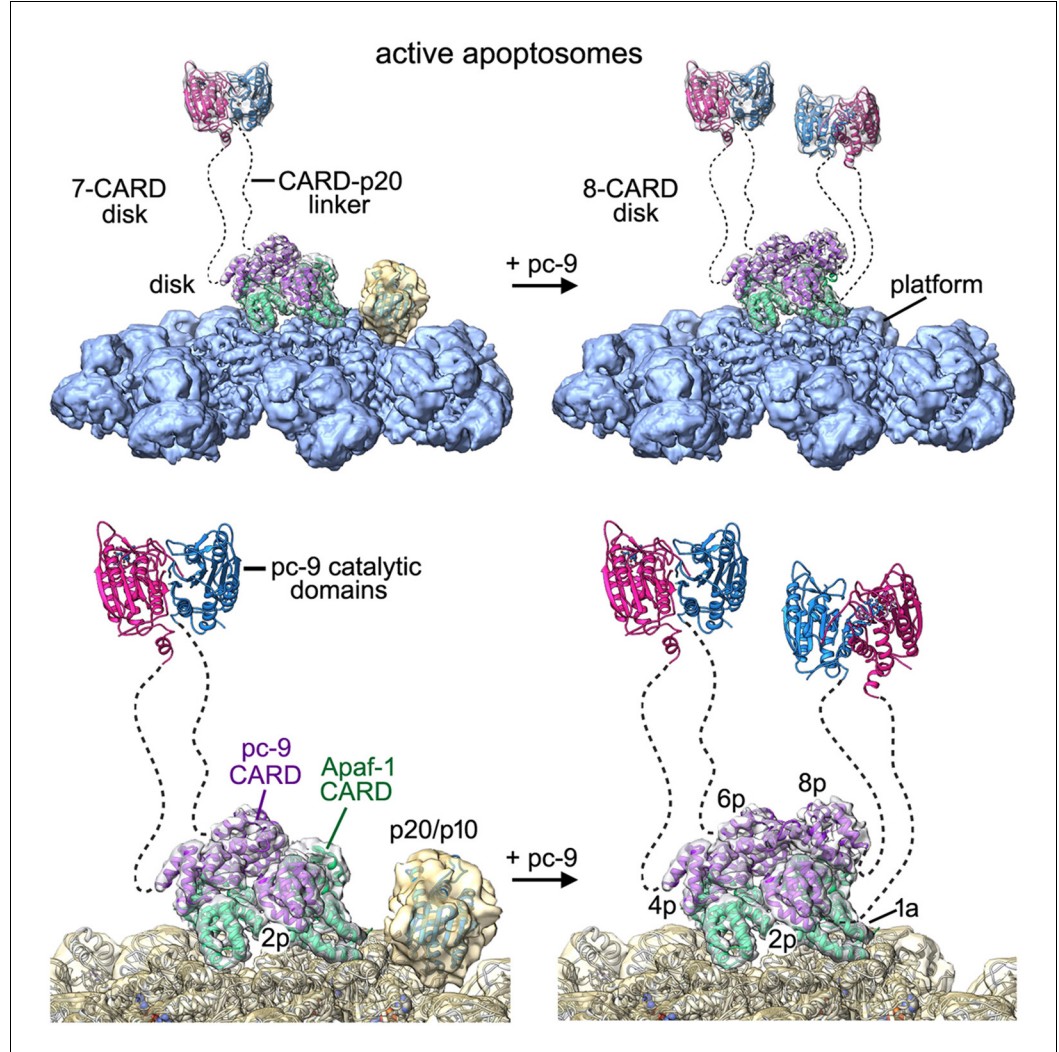

**Figure 9.** Models of the active apoptosome. In these models, procaspase-9 dimers (PDB 1JXQ) are flexibly-tethered to the CARD disk and may represent the activated enzyme. CARD-p20/p10 linkers are shown as dashed lines, and are drawn roughly to scale for their length. The number of possible pc-9 dimers in the active apoptosome may vary as a function of the number of pc-9 molecules tethered to the disk, with an odd number of zymogens creating parked catalytic domains for a single pc-9. For each panel, the top figure shows the entire active complex while the bottom figure shows a zoomed in view of the disk and central hub. **Left panels**: apoptosome with 3 bound pc-9 molecules and a 7 CARD disk; **right panels**: an active complex with 4 bound pc-9 molecules and an 8 CARD disk. Note that 4 Apaf-1 CARDs are always present in the CARD disk. For clarity the Apaf-1 CARD-NBD linkers are not shown.

central hub (this work, *Yuan et al., 2011a*). The two 'most favored' positions for parked pc-9 catalytic domains are next to the pc-9 CARD at position 2p in the disk. Our analysis further suggests that the acentric position of the disk may create the binding site for pc-9 catalytic domains. However, it is unlikely that adjacent parking sites would be occupied at the same time due to steric restraints. One implication of this finding is that parked catalytic domains could be attached to pc-9 CARDs located at all possible positions in the disk, due to the length of the CARD-p20 linker. Hence, the most favored parking sites are generally available to all four pc-9 molecules that may be tethered to the disk. We hypothesize that parked pc-9 catalytic domains may be held in reserve when an odd number of zymogens are bound to the apoptosome. This in turn, may prevent entanglement of linkers between parked pc-9 catalytic domains and flexibly-tethered pc-9 dimers anchored to the disk, with the latter representing active proteolytic centers. There is no ordered density in our map for other

pc-9 catalytic domains, and these domains can be released by a single clip in the CARD-p20 linker (*Yuan et al., 2010*). Thus, we surmise that the remaining pc-9 catalytic domains are flexibly-tethered to the disk through CARD-p20 linkers.

We propose that the active apoptosome is a dynamic proteolytic machine with flexibly-tethered pc-9 catalytic domains, whose nature may depend upon the number of pc-9 molecules that are bound at any instant. For example, a flexibly-tethered pc-9 dimer may form when three pc-9 molecules are anchored to form a 7 CARD disk, with the pc-9 catalytic domains that are odd man out occupying a parking spot on the central hub (*Figure 9*, left). However, if four pc-9 molecules are bound to the apoptosome they may form an 8-CARD disk with two flexibly-tethered pc-9 dimers, while leaving open parking spots on the central hub (*Figure 9*, right). By extension, if 5 pc-9 molecules were bound to the apoptosome then one pc-9 molecule might interact with an Apaf-1 CARD that is not incorporated into the disk. In this case, two flexibly-tethered pc-9 dimers may be formed on the apoptosome with an 8 CARD disk and a parked p20/p10 domain would also be present (e.g. *Figure 8—figure supplement 1B,C*). Since the pc-9 CARD at position 8p in the disk is present at lower occupancy, it is clear that active apoptosomes with three bound pc-9 molecules were formed with a higher likelihood under our in vitro conditions. At the same time, the high protein concentration in our experiments may have aided the formation of an 8 CARD disk in particles with four pc-9 molecules, whereas apoptosomes with a 7 CARD disk and three pc-9 molecules may be favored in vivo. Moreover, the KCl concentration may affect the stability of the pc-9 CARD at position 8p. The lower KCl concentration used in Buffer A, relative to physiological conditions (20 mM versus ~150 mM), may favor electrostatic interactions that predominate at CARD-CARD interfaces. Hence, the transient association of a pc-9 CARD at position 8p may be more favorable in low salt buffer. Also note that it remains an open question whether each putative pc-9 dimer on the active apoptosome may contain one (*Renatus et al., 2001*) or possibly two active sites.

Activation of pc-9 on the apoptosome has been proposed to follow the dictates of a molecular timer. In this model, auto-processing of pc-9 on the apoptosome triggers activation and subsequent release via exchange with unprocessed pc-9 in solution (*Malladi et al., 2009*). The rapid nature of the auto-proteolysis reaction suggests that two-chain pc-9 molecules, as used in our structure (*Figure 1—figure supplement 1A*), may account for most of the activity towards procaspase-3 (*Malladi et al., 2009*; *Hu et al., 2013*). Since there is no uncleaved pc-9 in our assembly reaction, we are not able to comment directly on the proposed turnover of pc-9 in the molecular timer hypothesis. However, the disposition of pc-9 on the apoptosome is similar for both unprocessed and cleaved pc-9 molecules. Indeed, a 3D map of apoptosomes with bound pc-9 CARDs, made by thrombinolysis of a single chain pc-9 with a triple point mutation (E306A/D315A/D330A), also showed the disk, while flexibly-tethered catalytic domains were released from the complexes (*Yuan et al., 2010*). Our structure of the active apoptosome suggests that flexibly-bound pc-9 catalytic domains are probably not able to affect the stability of the CARD-CARD disk. However, parked p20/p10 subunits of pc-9 could exert some effect on disk stability, to modulate the binding and release of pc-9, although this is speculative. Intriguingly, pc-9 CARDs are located on the top surface of the disk, which would provide a route for pc-9 molecules to dis-associate from the complex, such as occurs at position 8p. This step would down-regulate apoptosome activity until a replacement pc-9 is bound from solution. The major variable in evaluating possible turnover of bound pc-9 molecules is the overall stability of the CARD disk. Given the multiple interactions of pc-9 CARDs in positions 2p, 4p, and 6p, it appears that a CARD at position 8p will be the most likely to exchange, as observed in our analysis. Hence, the turnover of pc-9 at position 8p in the disk would lead to an oscillation in the number of potentially active pc-9 dimers bound to the apoptosome.

Finally, pc-9 molecules with shortened CARD-p20 linkers are not activated on the apoptosome as readily as wild type pc-9 (*Yuan et al., 2011a*). This suggests that the CARD-p20 linker may facilitate pc-9 activation by providing a suitable spacer between CARDs in the disk and the catalytic domains. Indeed, the requirement for a somewhat longer spacer may reflect the disposition of the four pc-9 CARD binding sites on the disk. Moreover, the CARD-p20 linker may also stabilize the pc-9 dimer when it is formed. In either case, a single clip in the linker releases the catalytic (p20/p10) domains from the apoptosome and results in a complete loss of proteolytic activity (*Yuan et al., 2010*, *2011a*). This argues that putative pc-9 dimers are only stable when present at a high local concentration on the apoptosome.

In summary, our structure of an active apoptosome suggests that most complexes will contain one (or possibly two) pc-9 dimers, which may be responsible for proteolytic activity and initiation of a cell death cascade. In a subsequent step, the cleavage and activation of procaspase-3 dimers may occur either in solution through a collisional process with flexibly-tethered pc-9 catalytic domains or the reaction may be facilitated by transient interactions with the Apaf-1 platform (*Yin et al., 2006*). Additional experiments are needed to address this point in light of recent experiments with chimeric platforms that contain N-terminal Apaf-1 CARDs, as these complexes are able to activate procaspase-9 and cleave procaspase-3 efficiently (*Hu et al., 2014*).

## Materials and methods

### Protein expression, purification and apoptosome assembly

His-tagged Apaf-1 was cloned into pFastBac, expressed in baculovirus infected sf9 cells and purified (*Acehan et al., 2002*; *Zou et al., 1999*). In brief, cells were harvested 48 hr post infection and lysed by homogenization in buffer T (20 mM tris pH 7.5, 50 mM NaCl, 1 mM PMSF, 1 mM benzamidine, 1 mM beta-mercaptoethanol). The lysate was centrifuged (100,000 g for 30 min) and the supernatant applied to a 2 ml Ni-NTA column; Apaf-1 was eluted in 250 mM Imidazole in Buffer T. Sample was dialyzed overnight at 4°C into 25 mM potassium phosphate (pH 7.5) with 50 mM NaCl. The dialyzate was applied to 2 ml of hydroxyapatite (BioRad HTP) in batch and eluted with 250 mM potassium phosphate (7.5) and 50 mM NaCl. The Apaf-1 was dialyzed into buffer A without $MgCl_2$ (10 mM Hepes pH 7.5, 20 mM KCl, 1 mM EDTA, 1 mM EGTA) and frozen at ~0.5 mg/ml until use. Clones for wild type and procaspase 9 mutants with a thrombin cleavage site in the CARD-p20 linker were expressed and purified as described (*Chao et al., 2005*; *Yuan et al., 2010*, *2011a*). To induce holo-apoptosome assembly, Apaf-1 (120 µg) in Buffer A without $MgCl_2$ was mixed with a slight excess of two chain pc-9 (~1.5x), bovine cytochrome c (~10 µg), 1 mM dATP, 1.5 mM $MgCl_2$ and incubated at 30°C for 5 min.

### Electron microscopy data acquisition

The holo-apoptosome was concentrated in Buffer A with an Amicon Ultra (10K cutoff) to ~4 mg/ml in a final volume of 30 µl. Concentrated sample (2.5 µl) was pipetted onto a C-Flat 300 mesh R1.2/1.3 holey grid (Protochips, Morrisville, North Carolina) that was freshly glow discharged and blotted for 1.5 s in 100% humidity at 20°C and plunge frozen in liquid ethane using a Vitrobot Mark 3 (FEI Company, Hillsboro, Oregon). Movie data were acquired on a Titan Krios electron microscope (FEI Company) operated at 300 kV, and equipped with a K2 Summit direct electron detector (Gatan, Pleasanton, California), a spherical aberration corrector and a Gatan Image Filter (GIF) with a slit width of ~20eV. Super-resolution counting mode was used for movie data collection with SerialEM (*Mastronarde, 2005*) at a nominal magnification of 81,000x, corresponding to 0.675 Å per super-resolution pixel, at a dose rate of ~10.2 electrons per physical pixel per second. Each total exposure of 40 electrons per $Å^2$ was fractionated into 18 frames and lasted 7.2 s. Defocus ranged from –1.5 to –2.4 µm (*Figure 1—source data 1*).

### Image processing

A total of 4900 movies were collected and stored in LZW compressed tiff format on the fly, with appropriate gain references. The data were uncompressed, corrected for the gain and binned 2X for processing with a script using IMOD (*Kremer et al., 1996*) giving a pixel size of 1.35 Å per pixel. The program UCSF MotionCorr was used to correct beam induced motions within each movie stack (*Li et al., 2013*), the first frame of each movie was discarded, and a summed micrograph (frames 2–18) was used for further processing. Contrast transfer function parameters were estimated by CTFFIND4 (*Rohou and Grigorieff, 2015*). In total, 134,970 particles were selected by automatic particle picking in RELION-1.3 (*Scheres, 2015*). A cleaned-up set of 92,867 particles was obtained by manual inspection, 2D and 3D classification in RELION-1.3 (*Scheres, 2012*). A previously published map (EMD-1931) was low pass filtered to 60 Å and used as the starting model for 3D classification. Refinement with c7 symmetry gave a map at 4.7 Å resolution (gold-standard $FSC_{0.143}$). Per particle corrections for beam-induced motion and B-factor weighting for radiation damage were carried out with statistical movie processing in RELION-1.3 (*Scheres, 2014*) with frames 2–18, a running average

of 5 frames and a standard deviation of 1 pixel for translations. A soft mask was used along with correction for the modulation transfer function of the K2 Summit detector to produce a map at 4.1 Å resolution, which was sharpened using automatic B-factor estimation (*Rosenthal and Henderson, 2003*) within RELION 1.3. Local resolution was estimated using ResMap (*Kucukelbir et al., 2014*).

A focused 3D classification was used to improve the local resolution of the sensor domain in a single Apaf-1 subunit as described (*Zhou et al., 2015*). In brief, the first Euler angle (α) from the alignment of each original particle image on the whole apoptosome was permutated cyclically by n times in 360°/7 increments, with n=1 to 6, to create six additional 'virtual particle' line entries in a new alignment data star file. In effect, this rotates each spoke density region within a given 2D image into the mask in the aligned 3D reference volume. The 3D mask covered the HD2, β-propellers, and cytochrome c in one Apaf-1 subunit. A focused 3D classification in RELION (*Scheres, 2012*) was done without an alignment search and members of the best class were then used to calculate an improved 3D map using relion_reconstruct, which places each particle projection only in its most probable orientation. After excluding the HD2, this gave a final sharpened map for the sensor domain with a resolution of 6.1 Å.

The CARD disk is rotationally blurred during c7 alignments, and refinement with c1 symmetry failed to clarify the map due to the small size (~75–80 kD) of the disk. Thus, we used a 3D classification that focused on the disk without local refinement. A soft mask was used that covered the disk and enclosed the NBD ring. In total, we identified seven classes with an asymmetric disk in different orientations, relative to the aligned c7 platform. After applying a soft mask to the disk, an estimated gold standard resolution of ~5.8 Å was obtained for each of the seven class maps. Sharpened 3D maps from each of the classes were then aligned and averaged in Chimera with vop_resample and equal weighting (*Pettersen et al., 2004*) to produce a final map of the disk.

The following steps were used to produce a map of the pc-9 apoptosome with c1 symmetry. Coordinates of shiny particles (92867) from RELION were used to extract particles from the movement corrected micrographs in EMAN2.1 (*Tang et al., 2007*) and CTFFIND4 parameters (*Rohou and Grigorieff, 2015*) were imported using a utility in EMAN2.1. After 2D classification with e2refine2d.py, particles in the best classes were selected (82091) and further segregated by a supervised 3D classification using e2refinemulti.py with two references. The first 3D reference was EMD-1931 (apoptosome with bound pc9 catalytic domains processed with c1 symmetry; *Yuan et al., 2011a*), while the second reference was constructed by removing density for the parked pc9 catalytic domains from the first reference map using Chimera. Particles in the class with bound catalytic domains (39985, ~48.7%) were used for c1 refinement with EMD-1931 as a reference, which was low pass filtered to 30 Å. This produced a map with a gold standard $FSC_{0.143}$ of 7.2 Å with α-helical rods resolved in the central hub (not shown). However, density for the bound pc-9 catalytic domains was not resolved at α-helical resolution presumably due to local flexibility.

## Model building

A previously published model containing the NBD, HD1, WHD, HD2 and β-propellers (PDB 3J2T; *Yuan et al., 2010*) was used for model building. The full Apaf-1 model was docked into the map as a rigid body in Chimera (*Pettersen et al., 2004*), followed by Molecular Dynamics Flexible Fitting (MDFF; *Trabuco et al., 2008*). The flexibly fitted model accounted well for most of the density, except for a loop between residues 349 to 364 in HD1. The resulting model was manually rebuilt and adjusted in Coot (*Emsley et al., 2010*) with intervening cycles of real space refinement in PHENIX to insure proper geometry (*Adams et al., 2010*). In the next stage, β-propellers were docked into the local map of the V-shaped domain using MDFF, while cytochrome c (PDB 3J2T), and individual CARDs from a crystal structure (*Hu et al., 2014*; PDB 4RHW) were docked with rigid body fitting in Chimera. We then calculated a model vs map $FSC_{0.5}$ to obtain an estimate of the final resolution for each of the models. All molecular figures were made with Chimera (*Pettersen et al., 2004*; *Goddard et al., 2005*) and Adobe Photoshop.

## Author information

Electron density maps and coordinates have been submitted to the Electron Microscopy Data Bank (EMD-8178) and the Protein Data Bank (5JUY), respectively.

## Acknowledgements

We thank Zhiheng Yu for help in collecting data at Janelia Farm and SJ Ludtke for helpful discussions. A cDNA plasmid for a chimeric Apaf-1 CARD/ClpP construct was provided by Dr. Yigong Shi. This work was supported by the National Institutes of Health grant GM063834.

## Additional information

### Funding

| Funder | Grant reference number | Author |
|---|---|---|
| National Institutes of Health | GM063834 | Christopher W Akey |

The funders had no role in study design, data collection and interpretation, or the decision to submit the work for publication.

### Author contributions

TCC, CWA, Conception and design, Acquisition of data, Analysis and interpretation of data, Drafting or revising the article; CH, Acquisition of data, Drafting or revising the article; IVA, Acquisition of data, Drafting or revising the article, Contributed unpublished essential data or reagents; SY, Drafting or revising the article, Contributed unpublished essential data or reagents

### Author ORCIDs

Christopher W Akey, http://orcid.org/0000-0002-3059-3121

## Additional files

### Major datasets

The following datasets were generated:

| Author(s) | Year | Dataset title | Dataset URL | Database, license, and accessibility information |
|---|---|---|---|---|
| Cheng TC, Hong C, Akey IV, Yuan S, Akey CW | 2016 | Active human apoptosome with procaspase-9 | http://www.rcsb.org/pdb/explore/explore.do?structureId=5JUY | Publicly available at the RCSB Protein Data Bank (accession no: 5JUY) |
| Cheng TC, Hong C, Akey IV, Yuan S, Akey CW | 2016 | Active human apoptosome with procaspase-9 | http://www.ebi.ac.uk/pdbe/entry/emdb/EMD-8178 | Publicly available at the the EMDataBank (accession no. EMD-8178) |

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
