## [Decision Letter]

Thank you for submitting your article "A near atomic structure of the active human apoptosome" for consideration by *eLife*. Your article has been reviewed by three peer reviewers, and the evaluation has been overseen by Sriram Subramaniam as Reviewing Editor and John Kuriyan as the Senior Editor. The following individuals involved in review of your submission have agreed to reveal their identity: Gabriel C Lander (Reviewer #2).

The reviewers have discussed the reviews with one another and the Reviewing Editor has drafted this decision to help you prepare a revised submission.

Summary:

Cheng et al. present the molecular architecture of the active form of the human apoptosome, based on a number of focused cryo-EM reconstructions that were determined from a single dataset. Using this structure, the authors describe the putative interactions that drive oligomerization of Apaf-1, the interaction cytochrome c with the sensor domain, and the spiraling CARD disk at the center of the apoptosome, which interestingly involves association of only 4 of the 7 CARDs. This new structure is coupled with prior apoptosome structures, as well as with native gradient gels to present a model for oligomerization, wherein cytochrome c binding induces conformational rearrangements that allow for nucleotide exchange and apoptosome assembly. The work of the authors significantly extends beyond their prior contributions to the analysis of the intricacies of apoptosome formation, and their model also provides novel insight into how caspase-9 may be activated on this signalling platform. The work also complements recent structural studies published by their international peers. One of the most striking findings of the authors is that their structural information might explain the current contradiction of how a supposedly heptavalent platform for caspase-9 activation is providing only a low number of active caspase-9 sites.

Overall the cryoEM analysis is well-done, and the authors show great expertise in their treatment of the data, using focused classification and other advanced image processing techniques to improve the resolution of disordered or flexible regions of the apoptosome complex. The overall quality of the density is consistent with the interpretation presented in the manuscript.

Essential revisions:

1) The "global" resolution is stated to be at 4.1 Angstroms resolution, yet a soft 3D mask was applied for this estimation – was this mask generated automatically, or manually, and more importantly, did this mask include the disordered sensor domains or CARD disk? If any structural features were cropped through the masking, then the reported "global" resolution is misleading.

2) The Resmap software is widely used by the EM community estimate local resolutions due to its ease of use. However, this software has repeatedly been shown to over-estimate resolutions. Comparison of the Resmap output with a similar program available in the Bsoft package is encouraged to ensure that the resolution claims are seen to be credible with map quality.

3) The PDB map associated with the combined EM density was generated by rigid body fitting previously determined crystal structures into the density, and these atomic models were refined into the density using a combination of MDFF and Phenix. For regions where side-chain densities are not clearly visible, the authors should limit the deposition to a C-α trace of the backbone.

4) Previous studies have shown that caspase-9 dimer activities towards procaspase-3 differ substantially from caspase-9 activities on apoptosomes. Can the structural model explain this phenomenon? In particular, can an allosteric contribution by the apoptosome backbone be excluded or substantiated? The authors also briefly touch on and somehow exclude the anti-dogmatic idea that caspase-9 monomers may be activated on the apoptosome backbone. It is not clear to me why that idea is or can be refuted. Would it be possible that both homodimers of active casp-9 can be formed on the apoptosome as well as active monomers? The authors should also explain better whether their structural model is consistent with the finding that the apoptosome can act as a molecular timer. Other than that, data presented in Figure 6 would probably be more convincing if the authors added a repeat in which a binding deficient PC9 was used as a negative control. Another suggestion is to comment on whether the data on cyt-c integration into the apoptosome and the contact regions (subsection “Sensor β-propellers and cytochrome c binding”, fourth paragraph) corresponds to cyt-c residues found to be critical for its proapoptotic potential (see e.g. published studies that compared yeast and human cyt-c).

5) The authors' structure of the platform of the pc-9 bound apoptosome and the unbound apoptosome recently reported by Zhou et al. 2015 is nearly identical (with the exception of the nucleotide orientation and whether Mg++ ions are involved in nucleotide binding), and for the most part the conclusions on the roles of cytochrome c and nucleotide binding in the formation of the apoptosome corroborate the findings of Zhou et.al. (2015). Perhaps more interesting is the increased resolution (~5.8 Angstroms) of the APAF-1:caspase-9 CARD disk and the confirmation of a previously postulated interaction of at least one catalytic caspase-9 P20/P10 dimer bound to the NBD ring of the central hub (Yuan et al. 2011). Previously, Akey's model of the apoptosome and caspase-9 (mostly from Yuan et al. 2010 and 2013) described the CARDs to be arranged in an acentric disk above the hub of a seven-spoked wheel of the apoptosome platform. The improved resolution of the disk reveals that it consists of 4 APAF-1 CARDS and 3 to 4 caspase-9 CARDS. Also, although their zipper like packing of APAF-1 and caspase-9 CARD domains differs from the crystal structure of the artificially created CARD disk of Hu et al. 2014, the author's structure is consistent with the existence of a second interface between the CARDs (dubbed type II interface) reported in that work. With one caspase-9 catalytic domain bound to the hub, this leaves the other two catalytic domains flexibly bound, potentially as an active dimer, though there is no evidence to verify this since the dimer is not resolved. The authors conclude that their model is consistent with the idea that the hub merely provides a platform for increasing the effective concentration of caspase-9 which drives dimerization and activation and that the caspase-9 catalytic domain is "parked" on the hub. However, their data in no way precludes that an as yet not understood allosteric mechanism accounts for the increased activity of apoptosome bound caspase-9. Indeed, Hu et al. 2015 identified mutations in APAF-1, K58E/K62E which are not found in any of the CARD:CARD interfaces and do not disturb apoptosome formation or recruitment of caspase-9, but severely interfere with the activity of the bound caspase-9. This suggests an allosteric mechanism exists for increased activation of apoptosome bound caspase-9. The author's only comment is that these mutations may affect binding of the CARD disc to the NBD hub. It would be helpful if they could elaborate on that further.

Other specific points:

In the first paragraph of the subsection “Domain and subunit interactions” – a list of residues potentially involved in lateral interactions in the central hub are listed. Are these based solely on the EM density? If so, the density should be included in a figure. Has mutagenesis been performed on these residues, or is there evidence of co-evolutionary variance to support these claims?

In the last paragraph of the subsection “Domain and subunit interactions” – some of the distances shown in Figure 3 are almost 5 Angstroms, which is beyond the range of hydrogen bonding. Due to the limited resolution of this region, it's possible that the modeling is inaccurate, so again, is there any other biochemical evidence supporting the role of the listed residues in stabilizing this region?

In the last paragraph of the subsection “Domain and subunit interactions” – The density described for the pi-cation interaction is questionable, if the authors want to include this statement, the density for this region should be shown in a figure.

In the third paragraph of the subsection “Sensor β-propellers and cytochrome c binding” – The authors state that all α-helices in cytochrome c are resolved. While the subunit can be unambiguously docked into the density, sausage-like densities that are consistent with α helices resolved to 6 Å resolution are not present.

In the fourth paragraph of the subsection “Sensor β-propellers and cytochrome c binding” – The interpretation of side-chain interactions based on docked crystal structures is appropriate, but discussion of visible Trp844 side chain density in a region where α helical densities are not discernible is over-interpretation.

In the third paragraph of the subsection “The CARD disk” – The authors mention that linker densities are visible, but the data is not shown. The symmetry mismatch in this region is a particularly interesting aspect of the manuscript, and the details of these linkers are relevant to understanding the mismatch – these linker densities should be shown. Despite their flexibility, is there any evidence of linker density for the linkers to the unassociated CARDs?

In the first paragraph of the subsection “A pc-9 catalytic domain on the hub” – Concerning the localization of the pc-9 catalytic domain – the authors' statement that EMAN2 uses a "density based approach in real space" needs further explanation. Is this simply cross-correlation-based projection matching? How is it different from other processing packages? The difference between the approach used in EMAN2 and RELION is not clear. Notably, a full description of the pc-9 processing is completely missing from the Methods section.

In the first paragraph of the subsection “A pc-9 catalytic domain on the hub” – Why is α-helical resolution reconstruction not shown? Do the p20 and p10 domains dock in with great accuracy into this reconstruction?

In the last paragraph of the subsection “A pc-9 catalytic domain on the hub” – How was alignment of composite map into the C1-symmetric pc-9-containing reconstruction performed? Is the cross correlation of this docked register substantially better than if it is docked into neighboring registers? Due to the speculative nature of this section, it's unclear what this paragraph adds to the manuscript.

---

## [Author Response]

*Essential revisions:*

*1) The "global" resolution is stated to be at 4.1 Angstroms resolution, yet a soft 3D mask was applied for this estimation – was this mask generated automatically, or manually, and more importantly, did this mask include the disordered sensor domains or CARD disk? If any structural features were cropped through the masking, then the reported "global" resolution is misleading.*

The estimation of 4.1Å for the global resolution used the entire map, prior to focused classification on the sensor domains and CARD disk. The mask was made in Relion based on user choice of threshold by visualizing the unsharpened map in Chimera. Please note that the CARD disk has been omitted for clarity in Figure 1—figure supplement 1, because it is rotationally-averaged and featureless at this stage of the analysis.

*2) The Resmap software is widely used by the EM community estimate local resolutions due to its ease of use. However, this software has repeatedly been shown to over-estimate resolutions. Comparison of the Resmap output with a similar program available in the Bsoft package is encouraged to ensure that the resolution claims are seen to be credible with map quality.*

We agree with the reviewers’ observation concerning the over-estimation of resolution by ResMap. As stated in the Methods section: “Local resolution was estimated using ResMap (Kucukelbir et al., 2014).” Please note that Resmap results are not presented as absolutes, but rather as a guide to highlight the inherent resolution gradient that is present. We have reinforced this point by changing the main text: “As estimated by ResMap (Kucukelbir et al., 2014) the central hub is at a nominal resolution of 3-4 Å, which is supported by the visibility of larger side chains, while the extended arm and peripheral sensor domains are in the range of 4.5-10 Å (Figure 1—figure supplement 1).”

*3) The PDB map associated with the combined EM density was generated by rigid body fitting previously determined crystal structures into the density, and these atomic models were refined into the density using a combination of MDFF and Phenix. For regions where side-chain densities are not clearly visible, the authors should limit the deposition to a C-α trace of the backbone.*

Well this is always a tricky point and in our opinion there is no single correct approach. Since we have used well defined crystal structures for regions of lower resolution, we feel that side chains should be left in place as a best estimate for that particular local region, and as a courtesy to biologists and structural groups who may use the pdb files in future. In particular, it is clearly indicated in the paper which regions are supported by higher resolution and the map itself serves as a clear guide. Side chains can easily be turned off in graphics programs, but having them present as they are found in the crystal structure allows one to more readily understand details of the local fold and packing, that might also apply within the larger ensemble. While contact regions between domains that form novel interactions, such as those between the two β-propellers will differ between the crystal structure and the ensemble, it is still more useful to have the residues present to get a general idea of the chemistry of a local environment. In addition, side chains help to guide the refinement of the protein backbone into the density with MDFF, so it seems reasonable to leave them in the final pdb file. Another approach would be to have two depositions, one in the Protein Data Bank with rather more strict guidelines pertaining to this issue, and a second repository for models, wherein a slightly more liberal, and in our opinion, more useful deposition can be made. However, for the moment this issue is left at the discretion of the authors.

*4) Previous studies have shown that caspase-9 dimer activities towards procaspase-3 differ substantially from caspase-9 activities on apoptosomes. Can the structural model explain this phenomenon? In particular, can an allosteric contribution by the apoptosome backbone be excluded or substantiated?*

We presume that this point is in reference to work from Hao Wu’s group showing that pc-9 linked to the C-terminus of a sequence that forms a Gcn leucine zipper is not as efficient as the pc-9 apoptosome at activating pc-3. That is a good point, but the story is complicated since the time course of pc-3 activation is not linear but actually slows significantly as activated caspase-3 accumulates, due to a reaction that mimics classic feedback inhibition by product, but the mechanism remains obscure (Yuan et al. 2011). We decided not to discuss pc-3 activation in great detail in this paper, since the paper focuses on pc-9. Future work may address this issue.

However, we added a final statement for the paper: “In a subsequent step, the cleavage and activation of procaspase-3 dimers may occur either in solution through a collisional process with flexibly-tethered pc-9 catalytic domains or the reaction may be facilitated by transient interactions with the Apaf-1 platform (Yin et al. 2006). Additional experiments are needed to address this point in light of recent experiments with chimeric platforms that contain N-terminal Apaf-1 CARDs, as these complexes are able to activate procaspase-9 and cleave procaspase-3 efficiently (Hu et al., 2014).”

Also see below.

*The authors also briefly touch on and somehow exclude the anti-dogmatic idea that caspase-9 monomers may be activated on the apoptosome backbone. It is not clear to me why that idea is or can be refuted. Would it be possible that both homodimers of active casp-9 can be formed on the apoptosome as well as active monomers?*

Of course it is theoretically possible that monomers of pc-9 might be activated on the apoptosome, as well as dimers. To test this idea, we obtained a clone of chimeric Apaf-1 CARD/ClpP molecules from Dr. Yigong Shi, as described below and in the main text of the paper, to investigate this possibility. We undertook this step because we had previously observed a single pc-9 catalytic domain on the platform of the holo-apoptosome and suggested that this might be involved in activation (Yuan et al., 2011).

“[…]heptamers, respectively, are able to activate pc-9 to a similar extent as the Apaf-1 apoptosome using a fluorescence-based proteolysis assay and by monitoring procaspase-3 cleavage (Hu et al., 2014). […] However, these studies do not rule out the possibility that procaspase-3 may interact with Apaf-1 at some point either during or after its activation (Yuan et al., 2011a and references therein).”

*The authors should also explain better whether their structural model is consistent with the finding that the apoptosome can act as a molecular timer.*

This is a complicated issue, however, we have now addressed this point. The paragraph below has been added to the Discussion.

“Activation of pc-9 on the apoptosome has been proposed to follow the dictates of a molecular timer. […] Hence, the turnover of pc-9 at position 8p in the disk would lead to an oscillation in the number of potentially active pc-9 dimers bound to the apoptosome.”

We also added the following point to the preceding paragraph concerning the stability of the pc-9 CARD at position 8p. “Moreover, the KCl concentration may affect the stability of the pc-9 CARD at position 8p. The lower KCl concentration used in Buffer A, relative to physiological conditions (20 mM versus ~150 mM), may favor electrostatic interactions that predominate at CARD-CARD interfaces. Hence, the transient association of a pc-9 CARD at position 8p may be more favorable in low salt buffer.”

*Other than that, data presented in Figure 6 would probably be more convincing if the authors added a repeat in which a binding deficient PC9 was used as a negative control.*

A good idea, but there are unfortunately no such mutants known to us, that might not be structurally compromised. Currently, we do not have an accurate enough docking of the catalytic domains to the platform region, due to the low resolution of this feature, to pin-point precise sites for mutation. This is a future goal of course.

*Another suggestion is to comment on whether the data on cyt-c integration into the apoptosome and the contact regions (subsection “Sensor β-propellers and cytochrome c binding”, fourth paragraph) corresponds to cyt-c residues found to be critical for its proapoptotic potential (see e.g. published studies that compared yeast and human cyt-c).*

Given the more limited α-helical resolution in this region of the map, we felt that space in the manuscript should be used to address other issues. However, we cite studies on cytochrome c and comment: “In total, 7 lysines from cytochrome c are found in these contact regions that include: lysines 25, 27, 39, 55, 72, 73 and 79 (also see Yu et al., 2001). However, higher resolution will be needed to pin down the precise nature of the interactions (also see mutation studies in Zhou et al., 2015).”

*5) The authors' structure of the platform of the pc-9 bound apoptosome and the unbound apoptosome recently reported by Zhou et al. 2015 is nearly identical (with the exception of the nucleotide orientation and whether Mg++ ions are involved in nucleotide binding), and for the most part the conclusions on the roles of cytochrome c and nucleotide binding in the formation of the apoptosome corroborate the findings of Zhou et.al. (2015).*

We concur that the general structure of the Apaf-1 platform for ground state and active apoptosomes is very similar, as reflected in the rmsd between the two platforms, and as predicted by our earlier work (Yuan et al., 2010). Nevertheless, this is an important point and also provides a benchmark for the methodology.

*Perhaps more interesting is the increased resolution (~5.8 Angstroms) of the APAF-1:caspase-9 CARD disk and the confirmation of a previously postulated interaction of at least one catalytic caspase-9 P20/P10 dimer bound to the NBD ring of the central hub (Yuan et al. 2011). Previously, Akey's model of the apoptosome and caspase-9 (mostly from Yuan et al. 2010 and 2013) described the CARDs to be arranged in an acentric disk above the hub of a seven-spoked wheel of the apoptosome platform. The improved resolution of the disk reveals that it consists of 4 APAF-1 CARDS and 3 to 4 caspase-9 CARDS. Also, although their zipper like packing of APAF-1 and caspase-9 CARD domains differs from the crystal structure of the artificially created CARD disk of Hu et al. 2014, the author's structure is consistent with the existence of a second interface between the CARDs (dubbed type II interface) reported in that work. With one caspase-9 catalytic domain bound to the hub, this leaves the other two catalytic domains flexibly bound, potentially as an active dimer, though there is no evidence to verify this since the dimer is not resolved. The authors conclude that their model is consistent with the idea that the hub merely provides a platform for increasing the effective concentration of caspase-9 which drives dimerization and activation and that the caspase-9 catalytic domain is "parked" on the hub. However, their data in no way precludes that an as yet not understood allosteric mechanism accounts for the increased activity of apoptosome bound caspase-9.*

We agree with most of the comments above, but one has to also take into account the large body of data showing that *all* other caspases are active as dimers. Please see our second reply to point 4 above. Hence, at this stage, the most parsimonious model would be that flexibly-tethered pc-9 catalytic domains probably form dimers. Unfortunately, the resolution of the parked pc-9 p20/p10 domains does not allow us to discern whether it is in an active or inactive conformation and its lower resolution is consistent with it also being rather dynamic, coming on and off the central hub.

*Indeed, Hu et al. 2015 identified mutations in APAF-1, K58E/K62E which are not found in any of the CARD:CARD interfaces and do not disturb apoptosome formation or recruitment of caspase-9, but severely interfere with the activity of the bound caspase-9. This suggests an allosteric mechanism exists for increased activation of apoptosome bound caspase-9. The author's only comment is that these mutations may affect binding of the CARD disc to the NBD hub. It would be helpful if they could elaborate on that further.*

Please note that K58E/K62E mutations are located on the bottom surface of the CARD disk and are not positioned where they could interact with pc-9 catalytic domains. We have now clarified this point in the paper.

“Two additional Apaf-1 CARD mutations (K58E, K62E) were found to greatly diminish pc-9 activation and occur outside of interfaces I and II (Hu et al., 2014). […] However, these two point mutations suggest that proper disk formation may be a pre-requisite for proper activation of pc-9 zymogens.”

*Other specific points:*

*In the first paragraph of the subsection “Domain and subunit interactions” – a list of residues potentially involved in lateral interactions in the central hub are listed. Are these based solely on the EM density? If so, the density should be included in a figure. Has mutagenesis been performed on these residues, or is there evidence of co-evolutionary variance to support these claims?*

Apaf-1 like molecules that may form a similar heptameric ring are restricted to vertebrates with the level of *identity* being very high, 55% in zebra fish to 87% and higher in other vertebrates. Hence, this approach is not particularly enlightening, and there is no space in the paper to go into a more detailed analysis. Please note that we point out possible interactions that are present as a guide, no systematic mutations have been made in Apaf-1, although now with this structure and that from Zhou et al., 2015 in hand, this would be possible. However, the process is fairly laborious with the protein being made in insect cells from a baculovirus.

*In the last paragraph of the subsection “Domain and subunit interactions” – some of the distances shown in Figure 3 are almost 5 Angstroms, which is beyond the range of hydrogen bonding. Due to the limited resolution of this region, it's possible that the modeling is inaccurate, so again, is there any other biochemical evidence supporting the role of the listed residues in stabilizing this region?*

A good point, however something is holding this critical interfacial region together. We have rephrased the text as follows:

“This surface is formed in part by possible interactions between Ser356 and Ser358 in the HD1-WHD loop to Asp271 and Ser272 in an extended region, which follows strand β4 in the NBD (Figure 3; Figure 2—figure supplement 4). However, some of these interactions are in the 4.5-5 Å range and thus, could be mediated in part by water molecules that have not been resolved.”

Also added to legend Figure 3 > “Possible interactions are indicated by dashed lines.”

*In the last paragraph of the subsection “Domain and subunit interactions” – The density described for the pi-cation interaction is questionable, if the authors want to include this statement, the density for this region should be shown in a figure.*

We have deleted this minor point from the paper.

*In the third paragraph of the subsection “Sensor β-propellers and cytochrome c binding” – The authors state that all α-helices in cytochrome c are resolved. While the subunit can be unambiguously docked into the density, sausage-like densities that are consistent with α helices resolved to 6 Å resolution are not present.*

Cytochrome c in the sensor domain may not be at 6A resolution, but individual helices are visible in the map at higher thresholds (e.g. 7.0 rmsd in Coot). At this threshold, some flexible regions of cyt c are outside the density envelope, but all helices are enclosed in connected rods.

*In the fourth paragraph of the subsection “Sensor β-propellers and cytochrome c binding” – The interpretation of side-chain interactions based on docked crystal structures is appropriate, but discussion of visible Trp844 side chain density in a region where α helical densities are not discernible is over-interpretation.*

As noted above α-helices are resolved in cytochrome c, however we modified this statement: “In addition, a strong density at the base of the V-shaped cleft may arise from a tryptophan (Trp844) in the 7-blade β-propeller. Two lysine residues from cytochrome c (Lys72 and Lys79) are also present in this region.”

We believe that this is a reasonable statement, given that there is no other logical explanation for this density in the map.

*In the third paragraph of the subsection “The CARD disk” – The authors mention that linker densities are visible, but the data is not shown. The symmetry mismatch in this region is a particularly interesting aspect of the manuscript, and the details of these linkers are relevant to understanding the mismatch – these linker densities should be shown.*

We realize that this is a particularly interesting aspect of the structure, but the current resolution and perhaps an inherent flexibility, precluded us from trying to build linker models. However, we have now used the SEGGER tool in Chimera to extract the 4 Apaf-1 CARD-NBD linkers and these linkers are now clearly shown in Figure 7.

*Despite their flexibility, is there any evidence of linker density for the linkers to the unassociated CARDs?*

There is no obvious density for the flexibly-attached CARDs.

*In the first paragraph of the subsection “A pc-9 catalytic domain on the hub” – Concerning the localization of the pc-9 catalytic domain – the authors' statement that EMAN2 uses a "density based approach in real space" needs further explanation. Is this simply cross-correlation-based projection matching? How is it different from other processing packages? The difference between the approach used in EMAN2 and RELION is not clear.*

EMAN2 uses a real space cross-correlation approach in projection matching but as described below, a critical step is the supervised 3D classification, which allows one to identify particles with bound pc-9 catalytic domains, and these particles are then refined to obtain a structure. I think that the questions of differences between EMAN2 and Relion in analyzing this type of very difficult problem, are beyond the space restraints for the paper, and in fact we’d need to have Steve Ludtke and Sjors Scheres debate this head to head! My understanding is that the real space method is a bit more sensitive to lower resolution features, while Fourier space Methods implemented in Relion are optimized to focus on higher resolution features. I would note however, that it recently became clear to us that there is a way to do a 3d autorefine in Relion that uses two or more models, to in effect carry out a supervised 3D classification. We have not processed our data with this approach in Relion, nor have we tried limiting the resolution of the particles through a strong low pass filter.

*Notably, a full description of the pc-9 processing is completely missing from the Methods section.*

We have added the following paragraph to Methods for the data processing with c1 symmetry that produced a map with bound pc-9 catalytic domains, as had been observed previously in Yuan et al., 2011. The modified data processing path is described below.

“The following steps were used to produce a map of the pc-9 apoptosome with c1 symmetry. […] However, density for the bound pc-9 catalytic domains was not resolved at α-helical resolution presumably due to local flexibility.”

*In the first paragraph of the subsection “A pc-9 catalytic domain on the hub” – Why is α-helical resolution reconstruction not shown? Do the p20 and p10 domains dock in with great accuracy into this reconstruction?*

As stated in the Methods: “This produced a map with a gold standard FSC_0.143_ of 7.2 Å. However, density for the bound pc-9 catalytic domains was not resolved at α-helical resolution presumably due to local flexibility.” This point is also made in the sixth paragraph of the subsection “The CARD disk”.

*In the last paragraph of the subsection “A pc-9 catalytic domain on the hub” – How was alignment of composite map into the C1-symmetric pc-9-containing reconstruction performed?*

By cross-correlation in Chimera

*Is the cross correlation of this docked register substantially better than if it is docked into neighboring registers?*

The cross correlation is better for the position discussed in the paper (0.9126) versus 5 others which ranged from 0.900-0.904, while an adjacent position was nearly the same at 0.9122. We have clarified this point now in the paper. Note that the small range is due to the shared structure of the platform in both maps and the fact that the differences are focused on the tilt and acentric location of the disk.

“Intriguingly, the catalytic domain can be located in two possible positions that are adjacent to a pc-9 CARD at position 2p within the CARD disk. […] This proximity would allow pc-9 catalytic domains parked on the hub to be connected to CARDs at all positions in the disk due to the long CARD-p20 linker.”

*Due to the speculative nature of this section, it's unclear what this paragraph adds to the manuscript.*

With regards to the possible significance: determining the relative position of this docking site shows that the parked pc-9 is placed on the side of the disk that has been displaced laterally to reveal more of the central hub and puts the bound pc-9 catalytic domains near positions 2p, 4p and 8p. This may prove to be important when we dissect further the mechanism of pc-9 activation and procaspase-3 feedback inhibition, but there is no space to go into this at this stage of the work. However, we have amended the text as follows.

“Our analysis further suggests that the acentric position of the disk may create the binding site for pc-9 catalytic domains. […] Hence, the most favored parking sites are generally available to all four pc-9 molecules that may be tethered to the disk.”